# Arf1-mediated lipid metabolism sustains cancer cells and its ablation induces anti-tumor immune responses in mice

Guohao Wang[1], Junji Xu[2], Jiangsha Zhao[1], Weiqin Yin[1], Dayong Liu[1], WanJun Chen[2] & Steven X. Hou[1]*

Cancer stem cells (CSCs) may be responsible for treatment resistance, tumor metastasis, and disease recurrence. Here we demonstrate that the Arf1-mediated lipid metabolism sustains cells enriched with CSCs and its ablation induces anti-tumor immune responses in mice. Notably, Arf1 ablation in cancer cells induces mitochondrial defects, endoplasmic-reticulum stress, and the release of damage-associated molecular patterns (DAMPs), which recruit and activate dendritic cells (DCs) at tumor sites. The activated immune system finally elicits antitumor immune surveillance by stimulating T-cell infiltration and activation. Furthermore, TCGA data analysis shows an inverse correlation between Arf1 expression and T-cell infiltration and activation along with patient survival in various human cancers. Our results reveal that Arf1-pathway knockdown not only kills CSCs but also elicits a tumor-specific immune response that converts dying CSCs into a therapeutic vaccine, leading to durable benefits.

[1] The Basic Research Laboratory, Center for Cancer Research, National Cancer Institute at Frederick, National Institutes of Health, Frederick, MD 21702, USA. [2] Mucosal Immunology Section, Oral and Pharyngeal Cancer Branch, National Institute of Dental and Craniofacial Research, National Institutes of Health, Bethesda, MD 20892, USA. *email: hous@mail.nih.gov

Cancer stem cells (CSCs) are a subpopulation of cells in a tumor that are in a stem cell state and have stem cell characteristics. CSCs may be responsible for treatment resistance, tumor metastasis, disease recurrence, and eventually patient death[1–3]. The ultimate goal of CSC research is to identify pathways that selectively regulate CSCs and then target these pathways to eradicate CSCs. Accumulated evidence suggests that CSCs are metabolically unique. It was recently reported that leukemia stem cells (LSCs) isolated from de novo acute myeloid leukemia (AML) patients uniquely rely on amino acids for oxidative phosphorylation (OXPHOS) to survive[4]. We previously found that the COPI/Arf1-mediated lipolysis pathway selectively sustains stem cells and transforms stem cells in Drosophila, and that knockdown of this pathway kills stem cells through necrosis[5]. Muscle stem cells (satellite cells) rely on mitochondrial fatty acid oxidation (FAO) but switch to glycolytic metabolism when they progress toward more committed states[6]. Some CSC-enriched disseminated tumor cells also obtain energy from fatty acids delivered through the fatty acid receptor CD36 expressed on a subset of highly aggressive CSCs[7,8]. These data together suggest that targeting the unique metabolism of CSCs, such as by blocking amino acid or lipid metabolism, may be a promising general strategy for killing CSCs.

However, the plasticity of tumor cells is well known[1]; CSCs and non-CSCs are mutually convertible in response to different signals or microenvironments. Therefore, CSCs can be recreated as long as non-CSCs and the tumor microenvironment remain intact, and only killing CSCs is not sufficient to elicit tumor regression. In this study, we found that ablating Arf1 in mice disrupts their lipid metabolism and results in lipid droplet accumulation, which further causes metabolic stress, including mitochondrial defects and endoplasmic reticulum (ER) stress. This metabolic stress selectively kills cells enriched with CSCs through necrosis in mice. The dying CSCs release damage-associated molecular patterns (DAMPs), which activate dendritic cells (DCs). The activated DCs further enhance T-cell infiltration and activation to stimulate an anti-tumor immunity. Our results reveal that knockdown of the Arf1 pathway has multimodal functions, it not only kills CSCs but also elicits a tumor-specific immune response in which dying CSCs are converted into a therapeutic vaccine to attract and activate immune cells for destroying the bulk tumors and resulting in durable efficacy of the treatment.

## Results

**Arf1-regulated lipolysis selectively sustains CSCs in mice.** Arf1 is one of the most evolutionarily conserved genes between Drosophila and mouse, with an amino acid identity of 95.6% between the two species (Supplementary Fig. 1a). We generated Arf1 conditional knockout (CKO) mice and are using them to investigate how the COPI/Arf1-mediated lipolysis pathway regulates stem-cell and CSC survival in these animals (Supplementary Fig. 1b). Lgr5 (leucine-rich-repeat-containing G-protein-coupled receptor 5, also known as Gpr49) is a Wnt target gene, and an inducible Lgr5-Cre knock-in allele (Lgr5-EGFP-IRES-CreERT2) targets genes to stem cells of the small intestine, colon, and various other adult tissues and cancers[9]. The Lgr5-CreERT2/Apc[f/f] (Lgr5/Apc) mouse is the best-known mouse model of CSCs[10]. Lgr5-CreERT2 encodes a tamoxifen-inducible Cre recombinase[11] that is expressed in long-lived intestinal/colon stem cells, while adenomatous polyposis coli (Apc) is a tumor suppressor. Knockdown of Apc using Lgr5-CreERT2 results in colon stem cell tumors[9]. Like Lgr5-CreERT2, Axin2-CreER is also selectively expressed in intestinal stem cells, and we found that the knockdown of Arf1 by Axin2-CreER could effectively ablate

stem cells[12] (Fig. 1a). We further studied the effect of Arf1 ablation in eradicating CSCs in Lgr5-CreERT2/Arf1[f/f]/Apc[f/f] (Lgr5/Arf1/Apc) mice and found that knocking down Arf1 dramatically reduced the stem cell tumor number (Lgr5/Apc: 95.8 ± 17.8; Lgr5/Arf1/Apc: 48.4 ± 18.1) (Fig. 1b, c) and significantly extended the lifespan of the Lgr5/Apc mice (Fig. 1d).

In Drosophila, we previously found that the COPI/Arf1-lipolysis-β-oxidation pathway sustains stem cells[5]. Thus, it was possible that the Arf1 pathway regulates CSC metabolism through a similar pathway. To test this hypothesis, we first analyzed lipid droplets in mouse colon cancer CT26 and human liver cancer Huh-7 cells. We found that treatment with the Arf1 inhibitor Brefeldin A (BFA) or Golgicide A (GCA) dramatically increased the lipid droplet accumulation in both CT26 and Huh-7 cells compared with DMSO-treated control cells (Supplementary Fig. 1c, d). We then analyzed the mitochondrial structure in Arf1-depleted intestines using electron microscopy (EM) (Fig. 1e, f, and Supplementary Fig. 1e–m). Compared to Lgr5/Apc mice, the cells in the Lgr5/Arf1/Apc intestine showed abundant aberrant or irregular mitochondria, poor cristae, numerous vacuoles, degeneration, and necrosis. We further found that the oxygen consumption rates (OCRs) and glycolytic reserve were significantly lower in Arf1 knockdown (sh-Arf1) CT26 and B16-F10 cells compared with control (sh-Scram) cells (Supplementary Fig. 1n–p).

The liver is a particularly lipid-rich microenvironment. To investigate Arf1's function in the developing liver, we deleted Arf1 from the ventral foregut endoderm with Foxa3-Cre (Foxa3-Cre/Arf1[f/f])[13] and in differentiated liver cells with Albumin-Cre (Alb-Cre/Arf1[f/f]). The liver progenitor cells, or hepatoblasts, are bipotent progenitor cells capable of differentiating into hepatocytes or cholangiocytes, and thus express genes specific to hepatocytes (Albumin and α-Fetoprotein-AFP), cholangiocytes (Cytokeratin 7-CK7 and 19-CK19), and progenitor cells (c-kit, Hnf4α, Hnf6). The homeobox gene Hhex (Hex) is expressed in the ventral foregut endoderm at E8.5 and is required for the very early liver development. C/EBPα is a fundamental regulator of hepatocyte differentiation and maturation that controls the expression of multiple liver-specific transcriptional genes[14,15]. We performed immunofluorescence (IF) staining to determine the stage at which Arf1 was successfully deleted by the Cre-driven recombination, and to examine the effect of Arf1 deletion on liver development using cell-type-specific markers. We found that Arf1 deletion with Foxa3-Cre resulted in significantly underdeveloped livers with lethality occurring at around mouse embryonic day 15.5 (E15.5) (Supplementary Fig. 2a, b), while Arf1 deletion with Alb-Cre resulted in normal mice (Supplementary Fig. 2c–f). We further found that Arf1 deletion with Foxa3-Cre resulted in a significant reduction in the hepatoblast markers Hnf4α and Hnf6 (Supplementary Fig. 2g–i), but had negligible effects on Hhex and C/EBPα (Supplementary Fig. 2i), suggesting that Arf1 specifically regulates hepatoblasts. We further found that Arf1 deletion with Foxa3-Cre resulted in the lipid droplet accumulation (Supplementary Fig. 2j, k) and necrotic death of hepatoblasts (Supplementary Fig. 2l, m). We also checked expression of fatty acid synthase (FASN), fatty acid translocase CD36, and carnitine palmitoyltransferase 1A (CPT1A) and found that FASN and CD36 had no change but CPT1A was decreased in comparison with controls (Supplementary Fig. 2n, o). In Drosophila, the COPI/Arf1-lipolysis pathway selectively sustains stem cells. Knockdown of this pathway in Drosophila stem cells, but not in differentiated cells, disrupts lipid metabolism and causes lipid droplet accumulation and stem cell necrosis[5]. Progenitors, such as hepatoblasts, share many properties with stem cells. Our findings that the knockdown of Arf1 in developing liver with Foxa3-Cre resulted in lipid droplet accumulation and hepatoblast necrosis,

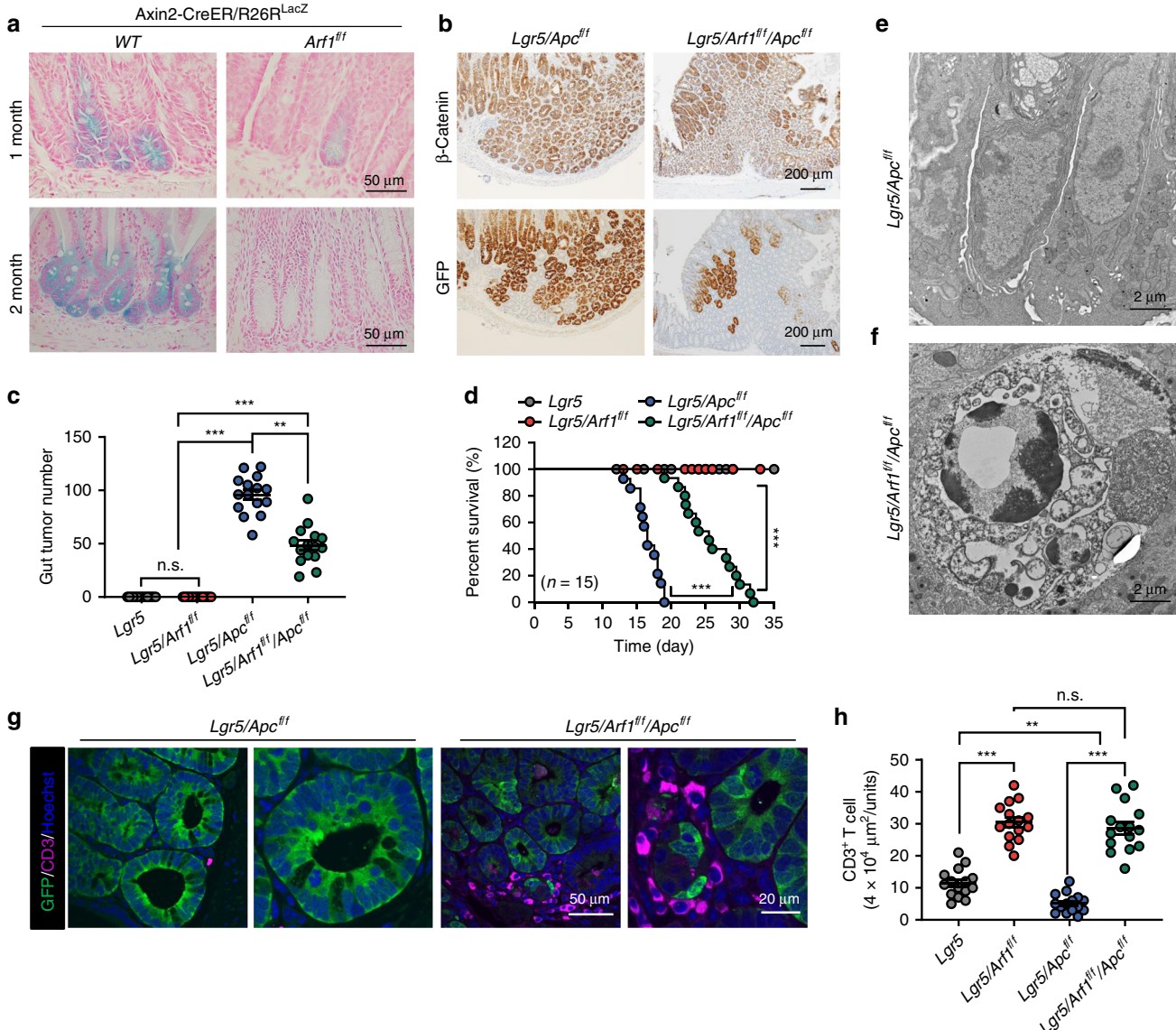

**Fig. 1 *Arf1* knockdown reduces the CSC and extends the lifespan of Lgr5/Apc mice. a** LacZ-stained sections of intestine from an Axin2-CreER/Rosa26R or an Axin2-CreER/Rosa26R/Arf1^f/f mouse that was treated with five intraperitoneal injections of tamoxifen to activate stem-cell-specific Cre and facilitate the loss of Arf1. **b** Lgr5/Apc or Lgr5/Apc/Arf1 mice were treated with three continued intraperitoneal injection of tamoxifen to activate the stem-cell-specific Cre and facilitate the loss of Apc and Arf1. β-Catenin and GFP mark stem cells. **c** Intestinal tumor number in the indicated genotypes ($n = 15$ mice per group; ***$p < 0.001$, $t$-test; repeated three independent times). **d** Percent survival curves of mice with the indicated genotypes ($n = 15$ mice per group; ***$p < 0.001$, $t$-test; repeated three independent times). **e, f** Electron microscopy sections of mouse intestinal crypts with the indicated genotypes. **g** Immunofluorescent staining of CD3, GFP, and Hoechst in intestinal sections of mice with the indicated genotypes. **h** Quantification of infiltrated CD3+ T cells in mice with the indicated genotypes ($n = 15$ mice per group; n.s. is not significant, **$p < 0.01$, ***$p < 0.001$, $t$-test; repeated three independent times). Each experiment replicate three times. Data are shown as the mean ± SEM. Scale bars are as indicated.

while the knockdown of *Arf1* in differentiated liver cells (with Alb-Cre) had no effect, suggested that Arf1 selectively sustains stem cells or progenitors in both *Drosophila* and mice. We further used the acetaminophen (APAP) to induce the liver injury in the Alb-Cre/Arf1^f/f mice and control (Alb-Cre/Arf1^f/+) mice. We found that the liver regeneration capacity was reduced once Arf1 was knocked out in hepatocytes in comparison with that of control mice (Supplementary Fig. 3a–c). It was reported that de-differetiation of hepatocytes was involved in injury-induced liver regeneration and Arf1 may play a role in that process.

Collectively, the above data suggest that: (1) the Arf1-regulated lipolysis pathway selectively sustains stem cells, progenitors, and cells enriched with CSCs in mice; and (2) disrupting the pathway

in these cells results in lipid droplet accumulation, mitochondrial defects, and cell necrosis.

**Arf1-KO induces anti-tumor immune responses in intestinal CSCs.** Necrosis is the main source of the damage signals in many tissue and tumor injuries that trigger sterile inflammation and anti-tumor immune responses[16–20]. We found a marked accumulation of CD3+, CD8+, and CD4+ cells in the intestine of Arf1-depleted mice (compare Lgr5/Arf1 with Lgr5 and Lgr5/Arf1/Apc with Lgr5/Apc) (Figs. 1g, h, 2a, b). We further studied the immune cell subsets in the Lgr5/Apc and Lgr5/Arf1/Apc mice by fluorescence-activated cell sorting (FACS) analysis (Fig. 2c–p). Among the gut antigen-presenting cells (APCs), we found that

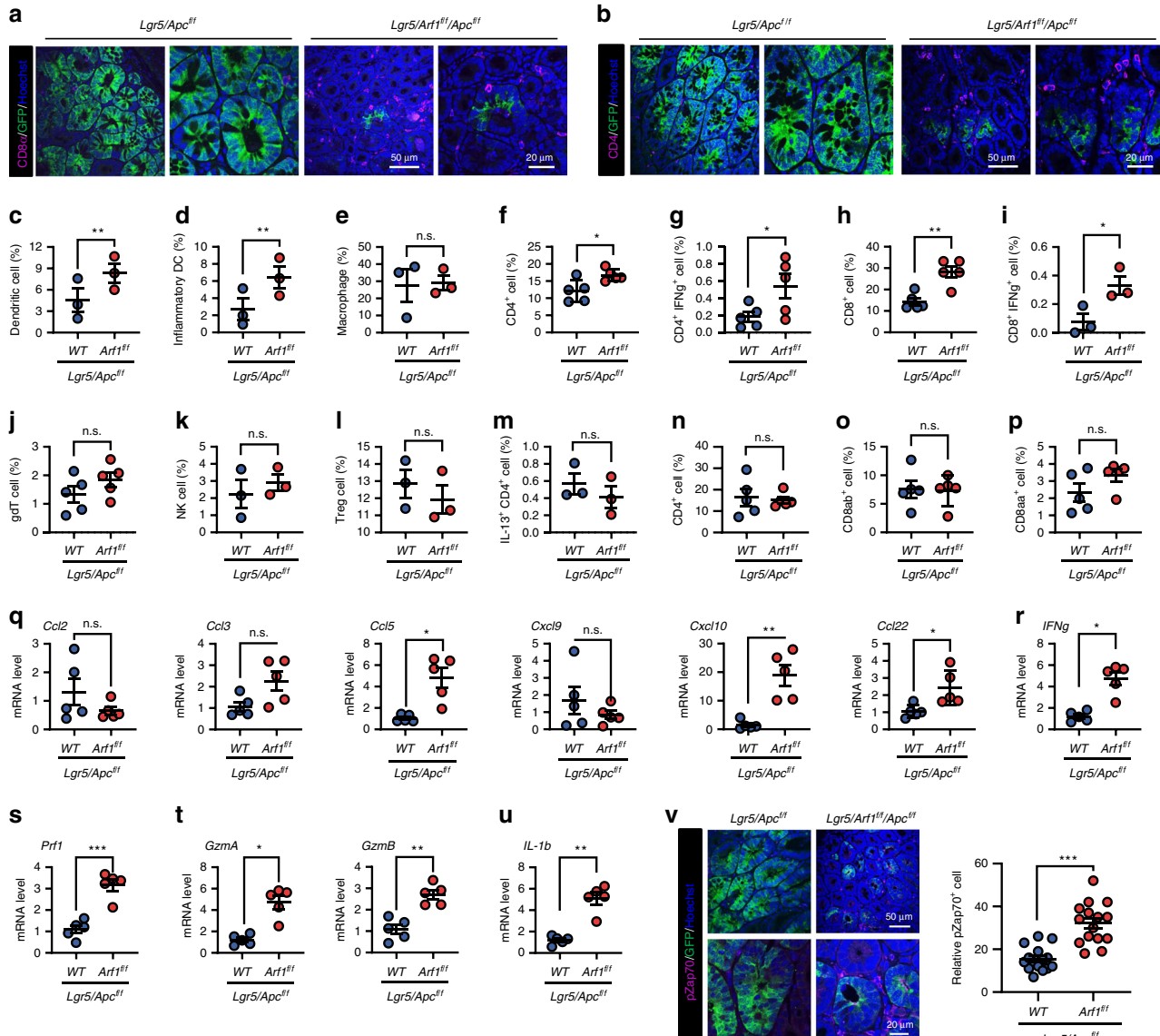

**Fig. 2 Arf1 ablation induces the infiltration and activation of immune cells in Lgr5/Apc mice. a, b** Immunofluorescent staining of GFP and CD8a (**a**) or CD4 (**b**) in the intestine of the indicated mice. **c–e** Flow cytometric analysis of gut APCs: DCs (**c**), inflammatory DCs (**d**), and macrophages (**e**) ($n = 3$ mice each group; **$p < 0.01$, $t$-test; repeat two independent experiments). **f–m** Flow cytometric analysis of the immune cells of LPLs: CD4+ T cells (**f**), IFNγ+CD4+ T cells (**g**), CD8+ T cells (**h**), IFNγ+CD8+ T cells (**i**), γδ T cells (**j**), NK cells (**k**), Treg cells (**l**), and IL-13+ CD4+ T cells, (**m**) ($n = 3$ or 5). **n–p** Flow cytometric analysis of the immune cells of IELs: CD4+ T cells (**n**), CD8αβ+ T cells (**o**), and CD8αα+ T cells (**p**) ($n = 5$ mice per group; *$p < 0.05$, **$p < 0.01$, $t$-test; repeat two independent experiments). **q–u** Relative gene expression of the indicated chemokines, cytokines, and granzymes ($n = 5$ mice each group, repeat two independent experiments). **v** Immunofluorescent staining of GFP and pZap70 in the intestine of the indicated mice ($n = 5$ mice each group; *$p < 0.05$, **$p < 0.01$, ***$p < 0.001$, $t$-test; repeat two independent experiments). Each experiment replicate two times. Data are shown as the mean ± SEM. Scale bars are as indicated.

DCs (Fig. 2c) and inflammatory DCs (Fig. 2d) were significantly increased in Arf1-depleted mice, while macrophages (Fig. 2e), neutrophils (Supplementary Fig. 4a), monocytes (Supplementary Fig. 4b), and B cells (Supplementary Fig. 4c) did not show a significant change. In the gastrointestinal (GI) tract, two types of lymphocytes are described, intraepithelial lymphocytes (IELs) and lamina propria lymphocytes (LPLs)[21]. Activated APCs migrate to secondary lymphoid tissues and draining lymph nodes (dNL) to prime naïve T cells and activated T cells; they then home to the LP (called LPLs) or infiltrate inflamed epithelium. Among the immune cells of the LPLs, we found that CD4+ T cells, IFNγ+ CD4+ T cells, CD8+ T cells, and IFNγ+CD8+ T cells significantly increased (Fig. 2f–i), while γδ T cells, NK cells,

Treg cells, IL-13+CD4+ T cells (Fig. 2j–m), Th17 cells, IL-13+γδT cells, IL-17+γδT cells, and TNFα+γδT cells (Supplementary Fig. 4d–g) did not significantly change. Among the immune cells of the IELs, we did not detect significant changes in CD4+ T cells, CD8αβ+ T cells, or CD8αα+ T cells (Fig. 2n–p). We further studied the immune cells in the spleen, dNL, and inguinal lymph node (iNL) and did not find any significant changes in the immune cells in these sites (Supplementary Fig. 4h–t).

We also found that some T-cell-related chemokines (CCL5, CXCL10, and CCL22) were upregulated in Lgr5/Arf1/Apc compared with Lgr5/Apc mice (Fig. 2q). The CCL5 and CXCL10 chemokines are reported to stimulate CD4+ and CD8+

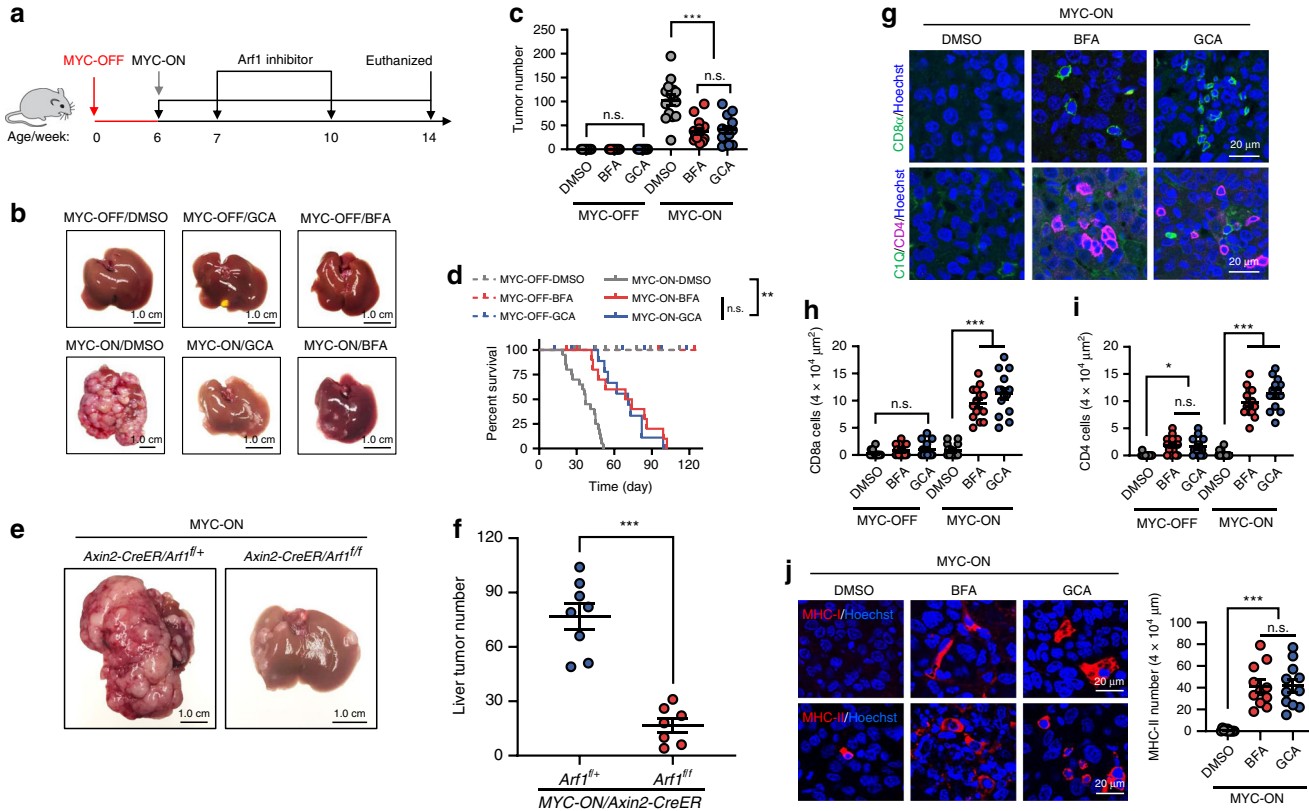

**Fig. 3 Arf1 inhibition induces anti-tumor immune responses in MYC-ON liver tumor mice. a** Experimental setup for the MYC-ON mice. **b** Representative liver images of MYC-OFF or MYC-ON mice treated with DMSO, GCA, or BFA. **c**, **d** Liver surface tumor counts (**c**) and survival curves (**d**) of mice with the indicated treatments ($n = 15$ mice each group; ***$p < 0.001$, $t$-test; pooled two independent experiments). **e** Representative liver images of MYC-ON/Axin2-CreER/Arf1$^{f/+}$ or MYC-ON/Axin2-CreER/Arf1$^{f/f}$ mice. **f** Liver surface tumor counts of **e** ($n = 8$ mice each group; ***$p < 0.001$, $t$-test; polled repeat two independent times). **g–i** Arf1 inhibition induces the expression of CD8 and CD4 in the liver of MYC-ON liver tumor mice ($n = 13$ mice each group; ***$p < 0.001$, $t$-test; polled repeat three independent times). **j** Arf1 inhibition induces the expression of MHC-I and MHC II in the liver of MYC-ON liver tumor mice ($n = 13$ mice each group; ***$p < 0.001$, $t$-test; polled repeat three independent times). Data are shown as the mean ± SEM. *$p < 0.05$, **$p < 0.01$, ***$p < 0.001$ by Student's $t$-test. Scale bars are as indicated.

lymphocyte migration in stromal compartments and inside tumors[22]. Furthermore, real-time PCR revealed elevated expressions of IFNγ, perforin, granzyme A (GzmA), granzyme B (GzmB), and IL-1β (Fig. 2r–u). IFNγ, perforin, GzmA, and GzmB are directly involved in the cytotoxic activities of T cells and are T-cell activation markers[23]. The activated DCs usually activate IFNγ-producing T cells through secreting IL-1β[24]. We also found that another T-cell activation marker, pZap70, was significantly elevated in the Lgr5/Arf1/Apc compared with Lgr5/Apc intestine (Fig. 2v). Collectively, these data suggest that the loss of Arf1 in Lgr5$^+$ stem cells triggers T-cell infiltration and activation, leading to CSC death and the prolonged survival of Lgr5/Arf1/Apc mice.

**Arf1-KO induces anti-tumor immune responses in liver tumor**. We further studied the anti-tumor effect of Arf1 inhibition in a mouse liver tumor model. Overexpressing the MYC proto-oncogene in adult mouse liver cells with the Tet system (Tet-o-MYC/LAP-tTA) by discontinuing doxycycline treatment reproducibly turns on MYC (MYC-ON) and induces a liver cancer that resembles hepatocellular carcinomas and/or hepatoblastomas[25]. We found that adding the Arf1 inhibitor GCA or BFA to the mouse food dramatically reduced the number of liver tumors (DMSO: 103.1 ± 41.1; BFA: 37.9 ± 24.4; GCA: 40.5 ± 28.4) (Fig. 3a–c, and Supplementary Fig. 5a) and significantly extended the lifespan of the MYC-ON mice (Fig. 3d).

Axin2-CreER is selectively expressed in one type of liver stem cells or Axin2$^+$ liver stem cells[8]. We found that knocking down Arf1 with Axin2-CreER could effectively ablate the stem cells (Supplementary Fig. 5b). We further studied the effects of Arf1 ablation by Axin2-CreER in the MYC-ON liver tumors (MYC-ON/Axin2-CreER/Arf1$^{f/f}$). We found that the selective depletion of Arf1 in Axin2$^+$ liver stem cells or CSCs with Axin2-CreER in the MYC-ON mice dramatically reduced the number of liver tumors (MYC-ON/Axin2-CreER/Arf1$^{f/+}$:76.8 ± 19.9; MYC-ON/Axin2-CreER/Arf1$^{f/f}$:16.7 ± 10.4) (Fig. 3e, f) and significantly extended the lifespan of the MYC-ON mice (Supplementary Fig. 5c). These data further suggest that Arf1 mainly functions in stem cells and CSCs, and that the anti-tumor effects of Arf1 ablation are due to the eradication of cells enriched with CSCs.

We also found a marked accumulation of CD4$^+$ and CD8$^+$ T cells and MHC-I and MHC-II molecules in the liver of MYC-ON mice treated with Arf1 inhibitors compared with DMSO-treated control mice (Fig. 3g–j). We also studied the immune cell subsets in the MYC-ON liver tumors of (MYC-ON/Axin2-CreER/Arf1$^{f/+}$) and (MYC-ON/Axin2-CreER/Arf1$^{f/f}$) mice by FACS analysis. We found that DCs, B cells, CD4$^+$ T cells, and CD8$^+$ T cells (Supplementary Fig. 5d) were significantly increased in the liver of Arf1-depleted MYC-ON mice, but the other immune cells did not change. We further found that a number of T-cell-related chemokines (CXCL10, CXCL11, and CCL22) were upregulated in the liver of MYC-ON mice treated with GCA compared with the DMSO-treated controls

(Supplementary Fig. 4u). The CXCL10 chemokine stimulates CD4[+] and CD8[+] lymphocyte infiltration into tumors[20] while CXCL10 and CXCL11 are T-cell-related chemokines. Furthermore, real-time PCR revealed elevated expressions of *IFNγ*, *perforin*, *GzmA*, *GzmB*, and *IL-1β* in the liver of mice treated with GCA compared with DMSO-treated controls (Supplementary Fig. 4v). Collectively, these data suggest that Arf1 inhibition or the loss of Arf1 in liver CSCs in MYC-ON mice triggered T-cell infiltration and activation, leading to liver tumor cell death and prolonged survival.

Our real-time PCR also revealed that the PD-L1 expression was significantly decreased in Lgr5/Arf1/Apc mice (Supplementary Fig. 4w), in the liver of MYO-ON mice treated with GCA compared with DMSO (Supplementary Fig. 4x), and in the tumor of BALB/c mice injected with murine colon carcinoma CT26 cells treated with GCA (Supplementary Fig. 4y) or with murine breast carcinoma 4T1 cells treated with GCA (Supplementary Fig. 4z). Moreover, we stained the tumor tissues with anti-PD-L1 antibody and found that the PD-L1 expressions in the tumor cell surface were reduced after Arf1 knockdown in comparison with those in control (Supplementary Fig. 6a–d). Because the PD-L1 can be regulated by HIF-1α, Myc, STATs, NF-κB, and AP-1[26,27], we checked expression of these transcription factors using qRT-PCR in the control and GCA-treated MYC-ON mice and found that only *AP-1/C-Jun* was decreased in GCA-treated MYC-ON mice in comparison with that in control mice (Supplemental Fig. 6e). Therefore, Arf1 inhibition may down-regulate PD-L1 through reducing AP-1/C-Jun signaling pathway. These data together indicate that PD-L1 reduction may be partially responsible for the activation of infiltrated T cells that enhances the anti-tumor effect of Arf1 ablation.

**Anti-tumor effect of Arf1-KO depends on CD4 and CD8 T cells**. To functionally confirm that the anti-tumor effects of Arf1-depletion in Lgr5/Apc and MYC-ON mice were indeed dependent on the observed adaptive immune response, we depleted the T lymphocytes after tumor formation by injecting anti-CD4 or anti-CD8 antibodies (Fig. 4). As expected, the antibody-mediated depletion of either CD4[+] or CD8[+] T cells impaired the Arf1-ablation-induced tumor suppression in both Lgr5/Apc and MYC-ON mice; the depletion of CD8[+] T cells had a stronger effect than the depletion of CD4[+] T cells, and the depletion of both CD8[+] and CD4[+] T cells had an additive effect. The tumor numbers (Lgr5/Apc: 77.8 ± 9.9 Lgr5/Arf1/Apc: 28.2 ± 7.8; CD4/Lgr5/Arf1/Apc: 42.2 ± 6.1; CD8/Lgr5/Arf1/Apc: 55.8 ± 15.2; CD4+CD8/Lgr5/Arf1/Apc: 71.2 ± 14.1; MYC-ON/DMSO: 89.6 ± 24.8; MYC-ON/GCA: 26.3 ± 18.6; MYC-ON/GCA/CD4: 51.0 ± 25.0; MYC-ON/GCA/CD8: 67.3 ± 22.6; MYC-ON/GCA/ CD8 + CD4: 87.5 ± 33.5) were dramatically increased (Fig. 4a, c–e), while the lifespans were significantly shortened in Arf1-ablated Lgr5/Apc and MYC-ON mice after the depletion of CD8[+] and CD4[+] T cells (Fig. 4b, f).

We further crossed Lgr5/Arf1/Apc mice with Rag1-KO and IFNγ-KO mice and did experiments with the Lgr5/Arf1/Apc/ Rag1-KO and Lgr5/Arf1/Apc/IFNγ-KO mice. In comparison with Lgr5/Apc and Lgr5/Arf1/Apc mice, we found that with Rag1 and IFNγ knockouts (particularly Rag1 knockout), the mice survival times were significantly decreased than those of Lgr5/ Arf1/Apc mice but were still somehow longer than those of Lgr5/ Apc mice. The tumor numbers in the Lgr5/Arf1/Apc/IFNγ-KO and Lgr5/Arf1/Apc/Rag1-KO also significantly increased in comparison with those in the Lgr5/Arf1/Apc mice (Lgr5/Apc: 99.6 ± 8.9; Lgr5/Arf1/Apc: 41.0 ± 5.6; Lgr5/Arf1/Apc/Rag1-KO: 90.4 ± 11.0; Lgr5/Arf1/Apc/IFNγ-KO: 73.0 ± 9.5) (Fig. 4g–i). Collectively, these data strongly suggest that the anti-tumor effects of

Arf1-ablation occur through the induction of a T-cell-dependent immune response.

**Arf1-KO induces DAMPs and DC infiltration**. The knockdown of Arf1 in stem cells and CSCs resulted in necrotic cell death and induced an adaptive immune response. Necrosis is the main source of damage signals in many tissue and tumor injuries and in anti-tumor immune responses;[17–20] therefore, we examined the DAMPs in Arf1-ablated systems. The most important DAMPs include: (i) the transport of the ER chaperone Calreticulin (Calr) to the cell surface; (ii) the secretion of ATP into the extracellular space; and (iii) the release of nuclear HMGB1 into the extracellular space. We first tested human liver cancer Huh-7 cells treated with the Arf1 inhibitor GCA. We treated the Huh-7 cells with either control DMSO or GCA and then injected the treated cells into athymic nude mice. After 3 weeks, the tumors from the treated cells were excised and stained for molecular markers. We found that Calr, HMGB1, another ER protein ERp46, and the lysosome protein LAMP1 were dramatically induced in the GCA-treated tumors (Fig. 5a, b). The nuclear HMGB1 moved into the extracellular space. Phosphorylated eIF2α (peIF2α) is an ER stress marker. ER stress specifically activates protein kinase R-like reticulum kinase (PERK) signaling, which phosphorylates eukaryotic initiation factor 2 alpha (eIF2α) to inhibit global protein translation[28]. ER stress is critical for instigating the danger signaling pathways responsible for the trafficking and emission of DAMPs that may act as danger signals[29,30]. We found that peIF2α was significantly increased in the GCA-treated tumors (Fig. 5c). Draper (Drpr) is a phagocytotic receptor in *Drosophila* that is involved in Arf1-deletion-induced stem cell necrosis[5,31]. We found that one of the mice Drpr homologs, PEAR1, was also significantly induced in the GCA-treated tumors (Fig. 5d). *Drosophila* Pretaporter (Prtp) is a ligand of the Drpr receptor[31]. ERp46 is a homolog of Prtp, suggesting that the ERp46 and PEAR1 pair is also involved in the Arf1-deletion-induced cell death in mice. We further found that cleaved and active Caspase 1 p20 but not cleaved Caspase 3 was also dramatically induced in GCA-treated tumors or in the Foxa3-Cre/Arf1[f/f] mice (Fig. 5e, Supplementary Fig. 7a). Because the cleaved and activated Caspase 1 is involved in an inflammasome-mediated pyroptosis, while cleaved Caspase 3 is involved in apoptosis[32], these data suggest that Arf1 inhibition may induce an inflammasome-mediated cell necrosis/pyroptosis. The direct treatment of Huh-7 or CT26 cells with Arf1 inhibitors induced the expression and translocation of HMGB1 and Calr (Fig. 5f, g, and Supplementary Fig. 7b). Furthermore, treating Huh-7 or CT26 cells with the Arf1 inhibitor BFA or GCA induced ATP secretion (Fig. 5h, i, and Supplementary Fig. 7c).

We also examined DAMPs and ER stress markers in the mouse intestinal CSC model. We found that both DAMPs (Calr and ERp46) and ER stress markers (BiP/GRP78, peIF2α, and CHOP) were significantly induced in the Arf1-ablated mice (Lgr5/Arf1 and Lgr5/Arf1/Apc) compared with Arf1-normal mice (Lgr5 and Lgr5/Apc) (Supplementary Fig. 7d–g). We further treated the MYC-ON mice with the Arf1 inhibitor BFA or GCA and found that DAMPs (Calr, ERp46, and HMGB1), ER stress markers (BiP, CHOP, and peIF2α), IL-1β, and the autophagy-related protein p62 were significantly induced in the inhibitor-treated compared with DMSO-treated mice (Supplementary Fig. 8a–l).

The surface-exposed Calr facilitates the engulfment of tumor-associated antigens by binding to LRP1/CD91 on DCs to initiate an immune response[28]. We found that LRP1 and the DC marker CD11c were dramatically induced in the intestine of Arf1-ablated mice (Lgr5/Arf1/Apc) compared with Arf1-normal mice (Lgr5/ Apc) (Fig. 5j, k), and in the liver of Arf1 inhibitor-treated MYC-

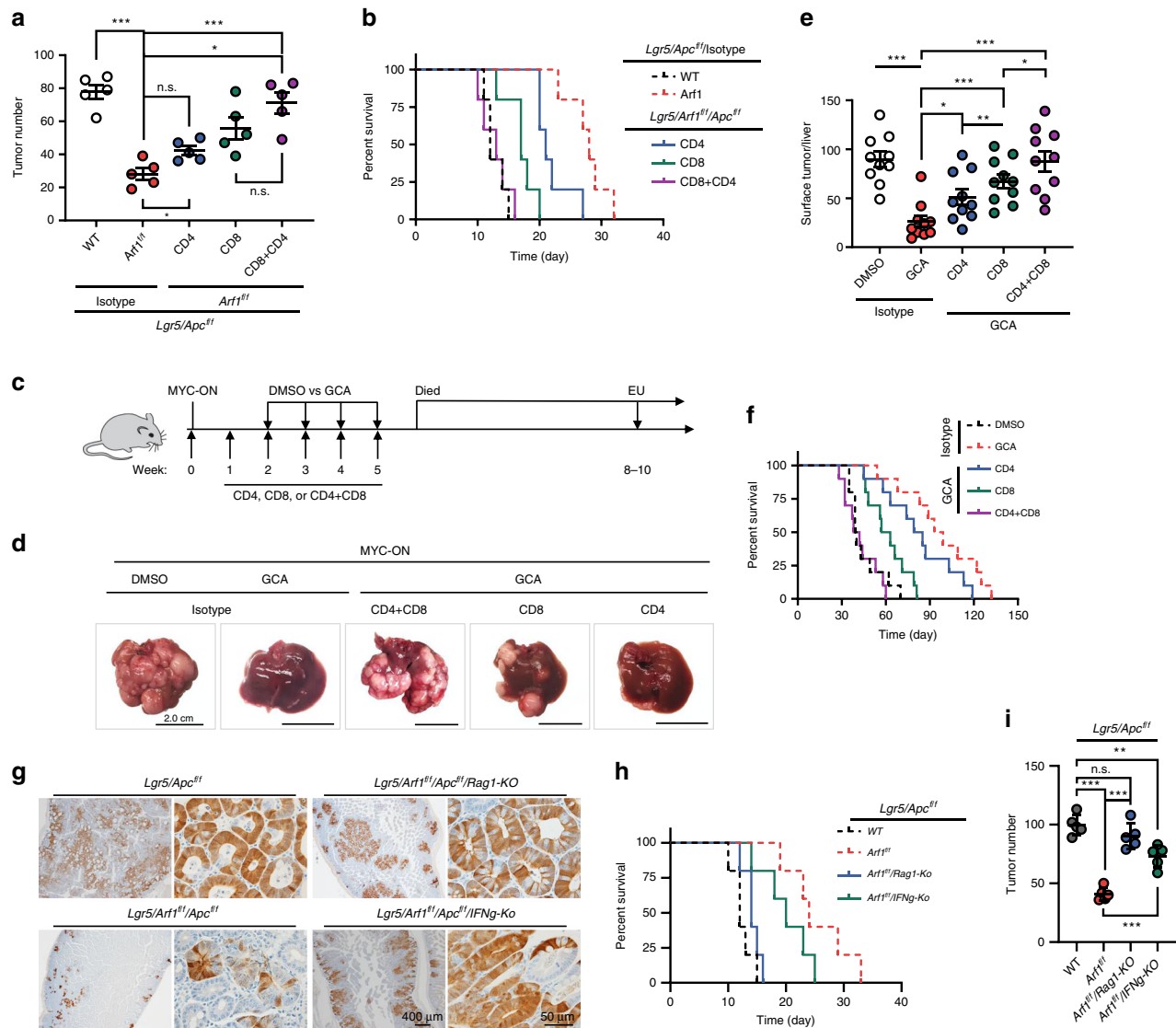

**Fig. 4 Prolonged survival of Lgr5/Apc and MYC-ON mice depends on the T cells. a** Intestinal tumor number in mice of the indicated genotypes treated with anti-CD4 (CD4) or/and anti-CD8 (CD8) antibodies. **b** Survival curves of mice with the indicated genotypes and treatments ($n = 5$ mice each group; *$p < 0.05$, **$p < 0.01$, ***$p < 0.001$, $t$-test; repeat two independent experiments). **c** Experimental setup for MYC-ON liver tumor mice treated with the indicated reagents. **d** Representative liver images of MYC-ON mice treated with the indicated reagents. Scale bars: 2.0 cm. **e** Liver surface tumor counts of MYC-ON treated with the indicated reagents ($n = 10$ mice per group; *$p < 0.05$, **$p < 0.01$, ***$p < 0.001$, $t$-test; pooled two independent experiments). **f** Survival curves of MYC-ON mice treated with the indicated reagents ($n = 10$ mice per group; *$p < 0.05$, **$p < 0.01$, ***$p < 0.001$, $t$-test; pooled two independent experiments). **g** Immunohistochemical staining of GFP in the indicated genotypes ($n = 5$ mice per group). **h–i** Percent of survival (**h**) and intestine tumor number (**i**) in the indicated mice ($n = 5$ mice each group; **$p < 0.01$, ***$p < 0.001$, $t$-test; repeat two independent experiments). Data are shown as the mean ± SEM. *$p < 0.05$, **$p < 0.01$, ***$p < 0.001$ by Student's $t$-test.

ON liver tumor mice compared with DMSO-treated mice (Fig. 5l). LRP1 and CD11c were colocalized in the Arf1 inhibitor-treated MYC-ON mouse liver (Fig. 5l), suggesting that Arf1 inhibition induced LRP1 in DCs.

Collectively, these data indicate that Arf1 inhibition triggers ER stress, and induces DAMPs and DC infiltration.

**Arf1-KO induces anti-tumor immune responses through DAMPs.** To examine whether DAMPs are essential for the anti-tumor immune response of Arf1 ablation, we first interfered with the action of ATP and HMGB1 in Lgr5/Arf1/Apc intestinal CSC mice by administering ARL61756 (ARL), oxATP, suramin, and the Toll-like receptor 4 (TLR4) inhibitor (Fig. 6a, b). ARL is an inhibitor of ecto-ATPases that enhances the action of ATP by artificially increasing the extracellular ATP concentration[33].

Oxidized ATP (oxATP) and suramin are purinergic receptor antagonists that inhibit ATP's action[34]. Extracellular HMGB1 elicits an immune response by binding to TLR4 on DCs and activates the TLR4-MYD88 pathway[35], so the TLR4 inhibitor should block HMGB1's function. We found that ARL had no effect on Lgr5/Arf1/Apc mice, but oxATP, suramin, and the TLR4 inhibitor significantly blocked the effect of Arf1 ablation. Under these treatments, the tumor numbers significantly increased (Lgr5/Apc: 101.2 ± 9.6; Lgr5/Arf1/Apc: 53.1 ± 7.7; ARL/Lgr5/Arf1/Apc: 50.2 ± 10.6; TLR4i/Lgr5/Arf1/Apc: 69.2 ± 11.7; OxATP/Lgr5/Arf1/Apc: 80.6 ± 10.5; OxATP/Lgr5/Arf1/Apc: 82.6 ± 7.0), and the mouse lifespan decreased, while the overall body weight did not change significantly, compared with mice without inhibitor treatment (Fig. 6b, c, and Supplementary Fig. 9a). We further tested these inhibitors in MYC-ON mice, and similarly found that

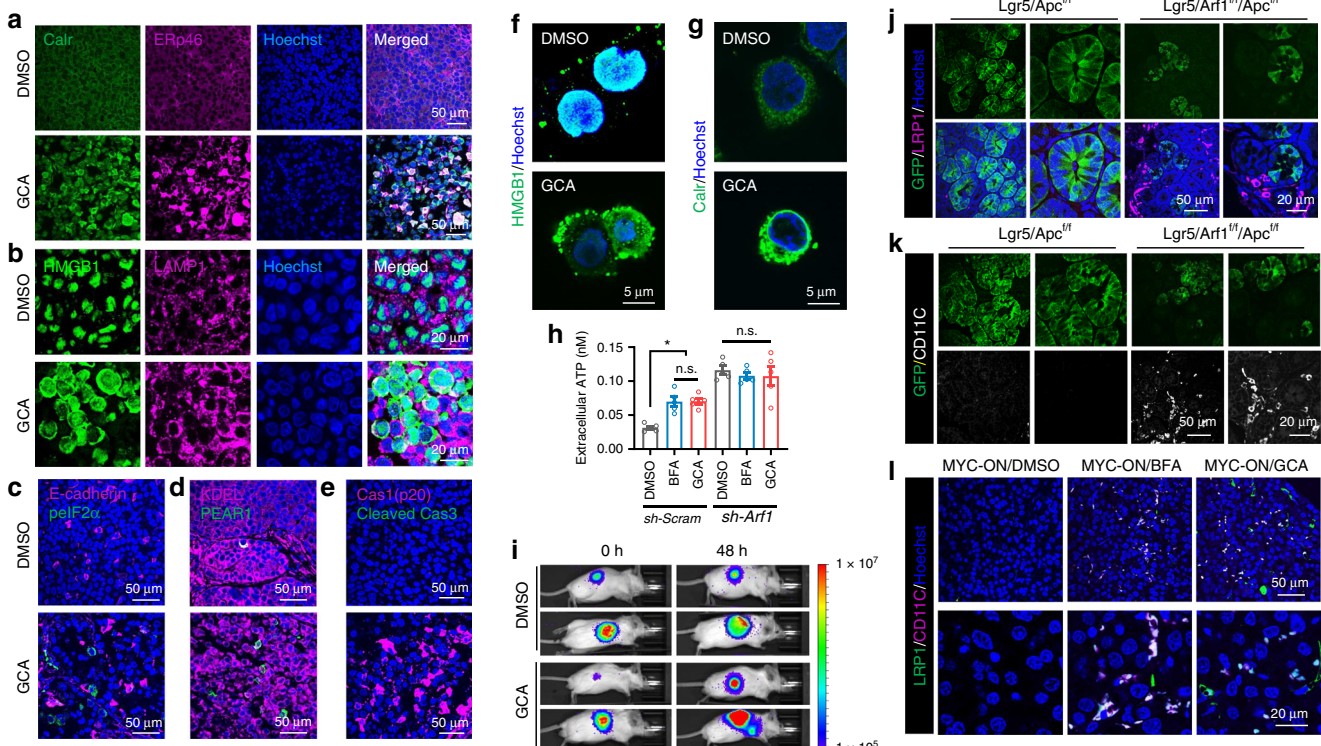

**Fig. 5 Arf1 inhibition triggers ER stress and induces DAMPs and DC infiltration. a–e** Huh-7 cells were treated with control DMSO or GCA and then injected into athymic nude mice. After 3 weeks, the tumors from the treated cells were excised and stained for molecular markers. Calr, HMGB1, ERp46, and the lysosome protein LAMP1 were dramatically induced in the GCA-treated tumors (**a** and **b**). Nuclear HMGB1 moved to the extracellular space (**b**). The ER stress marker peIF2α and the ERp46 receptor PEAR1 were also significantly increased in GCA-treated tumors (**c** and **d**). Cleaved and active Caspase 1 p20 but not cleaved Caspase 3 was also dramatically induced in the GCA-treated tumor (**e**). **f, g** Direct treatment of Huh-7 cells with GCA induced the expression and translocation of HMGB1 (**f**) and the translocation of Calr (**g**) from the nucleus to the extracellular space. **h** Treatment of Huh-7 cells with the Arf1 inhibitor BFA or GCA induced ATP secretion from control but not Arf1 (sh-Arf1) knockdown cells ($n = 5$ wells cell each group, $*p < 0.05$, $t$-test; repeat three independent experiments). **i** Treatment of mice with Arf1 inhibitor GCA increased ATP release in vivo. **j, k** LRP1 and CD11c expression in the intestine of Lgr5/Apc or Lgr5/Arf1/Apc mice. **l** LRP1 and CD11c expression in the liver of MYC-ON liver tumor mice treated with the indicated reagents. Each experiment replicates three times. Data are shown as the mean ± /SEM. $*p < 0.05$, $**p < 0.01$ by Student's $t$-test. Scale bars are as indicated.

that ARL had no effect on the Arf1 inhibitor GCA-treated mice, while oxATP, suramin, and the TLR4 inhibitor dramatically blocked the anti-tumor effect of GCA (Fig. 6d–f, and Supplementary Fig. 9b–d). Collectively, these data suggest that Arf1 ablation induces its anti-tumor effect through ATP and HMGB1.

We further knocked down Arf1, BiP, ERp46, HMGB1, and calr with specific small interfering RNAs (shRNAs) (Supplementary Figs. 11a–d, 12a, 13a). We transfected murine colon carcinoma CT26 cells, melanoma B16-F10 cells, and breast carcinoma 4T1 cells with the indicate combinations of shRNAs in lentiviral vectors. The CT26, B16-F10, and 4T1 cells contain 98.3%, 40.1%, and 76.6% of the CD133+CD44hi cells (potential CSC markers), respectively (Supplementary Fig. 9e). We first tested the transfected cells in athymic nude mice and found that Arf1 knockdown or inhibition did not affect the tumorigenic capability of B16-F10, CT26, or 4T1 cells in the immune-deficient mice (Supplementary Fig. 9f–h), suggesting that the anti-tumor activity of Arf1 knockdown does not occur through a directly cytotoxic effect on tumor cells. We then injected B16-F10 cells that were transfected with either sh-Arf1 or sh-Scram into wild-type and Rag1-knockout (Rag1-KO) (Fig. 6g, and Supplementary Fig. 10a) C57BL/6J mice or CT26 cells that were transfected with either sh-Arf1 or sh-Scram into wild-ype and INFγ knockout (INFγ-KO) (Fig. 6h, and Supplementary Fig. 10b) BABL/c mice. We found that the anti-tumor effect of Arf1 knockdown in wild-type mice was lost in both the Rag1-KO and INFγ-KO mice. We also found

that the anti-tumor effect of Arf1 knockdown in wild-type mice was lost in both the DCs-KO and P2RX7-KO mice (Fig. 6i, j, and Supplementary Fig. 10c, d). These data further suggest that Arf1 knockdown in tumor cells promotes anti-tumor responses by eliciting a potent DCs-ATP-INFγ-mediated anti-tumor T-cell immunity.

We also investigated the effect of Arf1 knockdown on tumor metastasis. We injected B16-F10 cells that were transfected with either sh-Arf1 (sh-Arf1-B16) or sh-Scram (sh-Scram-B16) into tail vein of C57BL/6J mice. We found that the tumors in sh-Scram-B16 mice metastasized to lung (Supplementary Fig. 10e, f), kidney (Supplementary Fig. 10g), and testis (Supplementary Fig. 10h) after 15 days of injection, while the tumors in sh-Arf1-B16 mice had negligible tumor metastasis. These data suggest that that Arf1 knockdown in tumor cells prevents tumor metastasis.

We further subcutaneously injected CT26 cells or B16-F10 cells that were either transfected with shRNAs or treated with the ER stress inhibitor 4-phenylbutyric acid (PBA) or GSK2606414 into BALB/c mice or C57BL/6J mice (Fig. 6k–o, Supplementary Fig. 11e–l and Supplementary Fig. 12). Tunicamycin can induce the ER stress, we also treated the MYC-ON mice with Tunicamycin plus BFA and found that co-treatment with the two inhibitors can further reduce the tumor number (MYC-ON/ DMSO: 63.0 ± 18.0; MYC-ON/BFA: 16.8 ± 7.5; MYC-ON/Tuni: 47.3 ± 19.8; MYC-ON/BFA+Tuni: 5.8 ± 5.0) (Supplementary Fig. 11m, n). Therefore, we found that: (1) the knockdown of

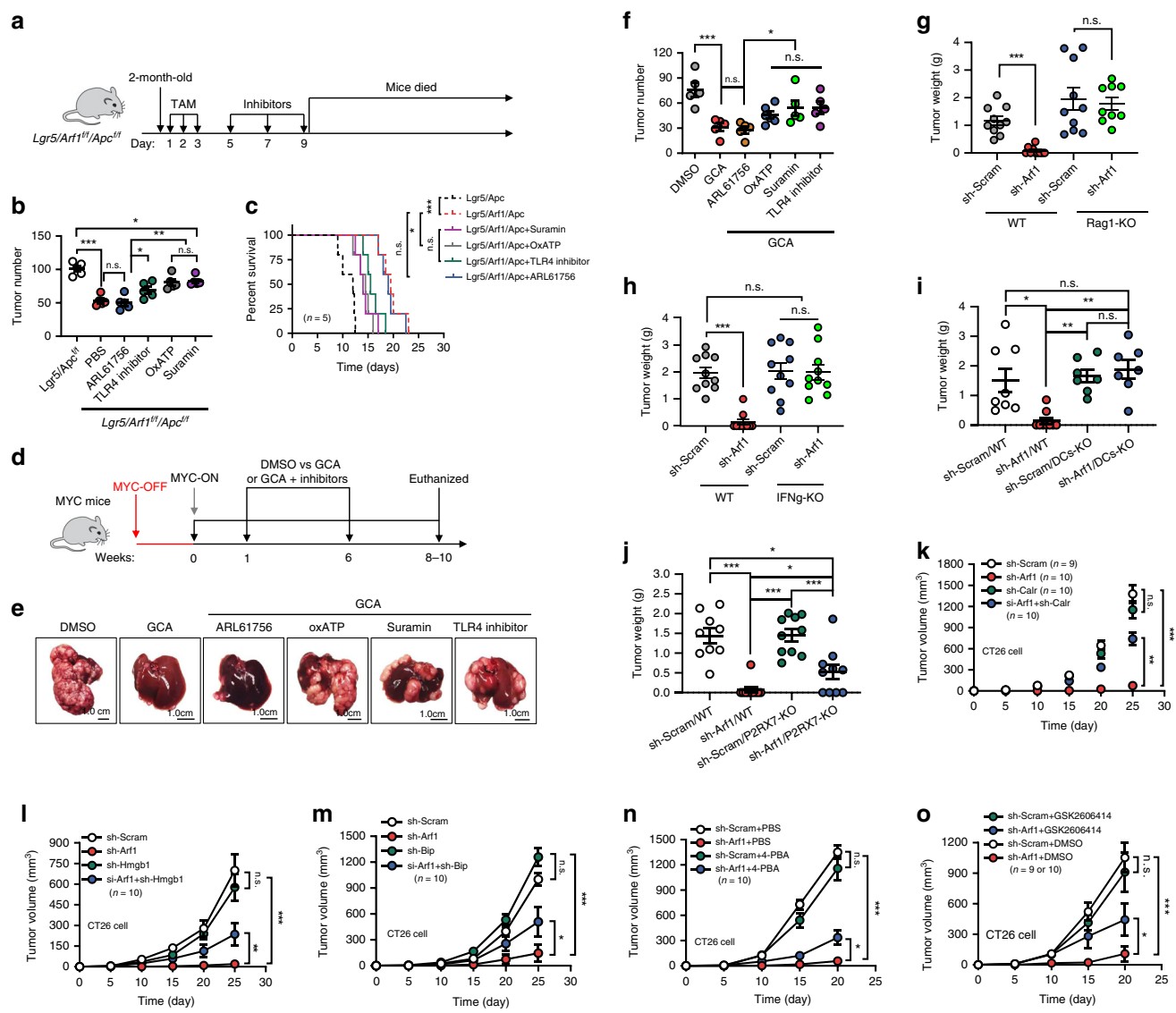

**Fig. 6 Arf1 ablation induces anti-tumor immune responses through DAMPs. a** Experimental setup for Lgr5/Arf1/Apc mice. TAM: injection of tamoxifen. **b** Tumor numbers in mice treated with the indicated inhibitors. **c** Survival curves of Lgr5/Arf1/Apc mice treated with the indicated reagents ($n = 5$ mice each group; *$p < 0.05$, **$p < 0.01$, ***$p < 0.001$, $t$-test; repeat two independent experiments). **d** Experimental setup for the MYC-ON liver tumor mice. **e** Representative liver images of MYC-ON mice treated with the indicated reagents. **f** Number of liver surface tumors in MYC-ON mice treated with the indicated inhibitors ($n = 5$ mice per group; *$p < 0.05$, ***$p < 0.001$, $t$-test; repeat two independent experiments). **g** Tumor weight of wild-type or Rag1-KO C57BL/6J mice receiving B16-F10 cells treated with the indicated sh-RNAs (sh-Scram and sh-Arf1 in WT group, sh-Scram in Rag-KO group, $n = 10$ mice each group; sh-Arf1 in Rag-KO group, $n = 9$ mice; ***$p < 0.001$, $t$-test; polled two independent experiments). **h** Tumor weight of wild-type or IFNγ-KO BALB/c mice receiving CT26 cells treated with the indicated sh-RNAs ($n = 10$ mice each group; *$p < 0.05$, $t$-test; repeat two independent experiments). **i** Tumor weight of wild-type or DCs-KO C57BL/6J mice receiving B16-F10 cells treated with the indicated sh-RNAs (sh-Scram/WT, $n = 8$ mice; sh-Arf1/WT, $n = 10$ mice; sh-Scram/DCs-KO and sh-Arf1/DCs-KO, $n = 7$ mice; *$p < 0.05$, **$p < 0.01$, ***$p < 0.001$, $t$-test; repeat two independent experiments). **j** Tumor weight of wild-type or P2RX7-KO C57BL/6J mice receiving B16-F10 cells treated with the indicated sh-RNAs (sh-Scram/WT, $n = 9$ mice; sh-Arf1/WT, sh-Scram/P2RX7-KO and sh-Arf1/P2RX7-KO, $n = 10$ mice each group; *$p < 0.05$, ***$p < 0.001$, $t$-test; repeat two independent experiments). **k–o** Tumor volume curves of BALB/c mice receiving transplants of murine colon carcinoma CT26 cells treated with the indicated sh-RNAs and inhibitors (**k**, sh-Scram $n = 9$ mice, other groups $n = 10$ mice; **l–n**, $n = 10$ mice per group; **o**, sh-Scram/DMSO and sh-Scram/GSK2606414, $n = 9$ mice per group; sh-Arf1/DMSO and sh-Arf1/GSK2606414, $n = 10$ mice per group; *$p < 0.05$, **$p < 0.01$, ***$p < 0.001$, $t$-test; repeat two independent experiments). Data are shown as the mean ± SEM. Scale bars are as indicated.

Arf1 dramatically blocked the tumorigenic capability of CT26 and B16-F10 cells; (2) the knockdown of Calr or HMGB1 or BiP significantly reduced the anti-tumor activity of Arf1 knockdown; (3) treatment with the ER stress inhibitor PBA or GSK2606414 significantly blocked the anti-tumor activity of Arf1 knockdown while treatment with ER stress inducer Tunicamycin significantly enhanced the anti-tumor activity of Arf1 knockdown; and (4) the knockdown of ERp46

did not significantly change the anti-tumor activity of Arf1 knockdown.

We also investigated the effect of Arf1 inhibition in CT26 cells on the production of IL-1β by DCs and INFγ by CD8⁺ T cells in co-culture experiments (Supplementary Fig. 13a–l). Consistent with the above findings, we found that Arf1 inhibition either with GCA or sh-Arf1 in CT26 cells dramatically induced the production of IL-1β and INFγ, and that the knockdown of Calr,

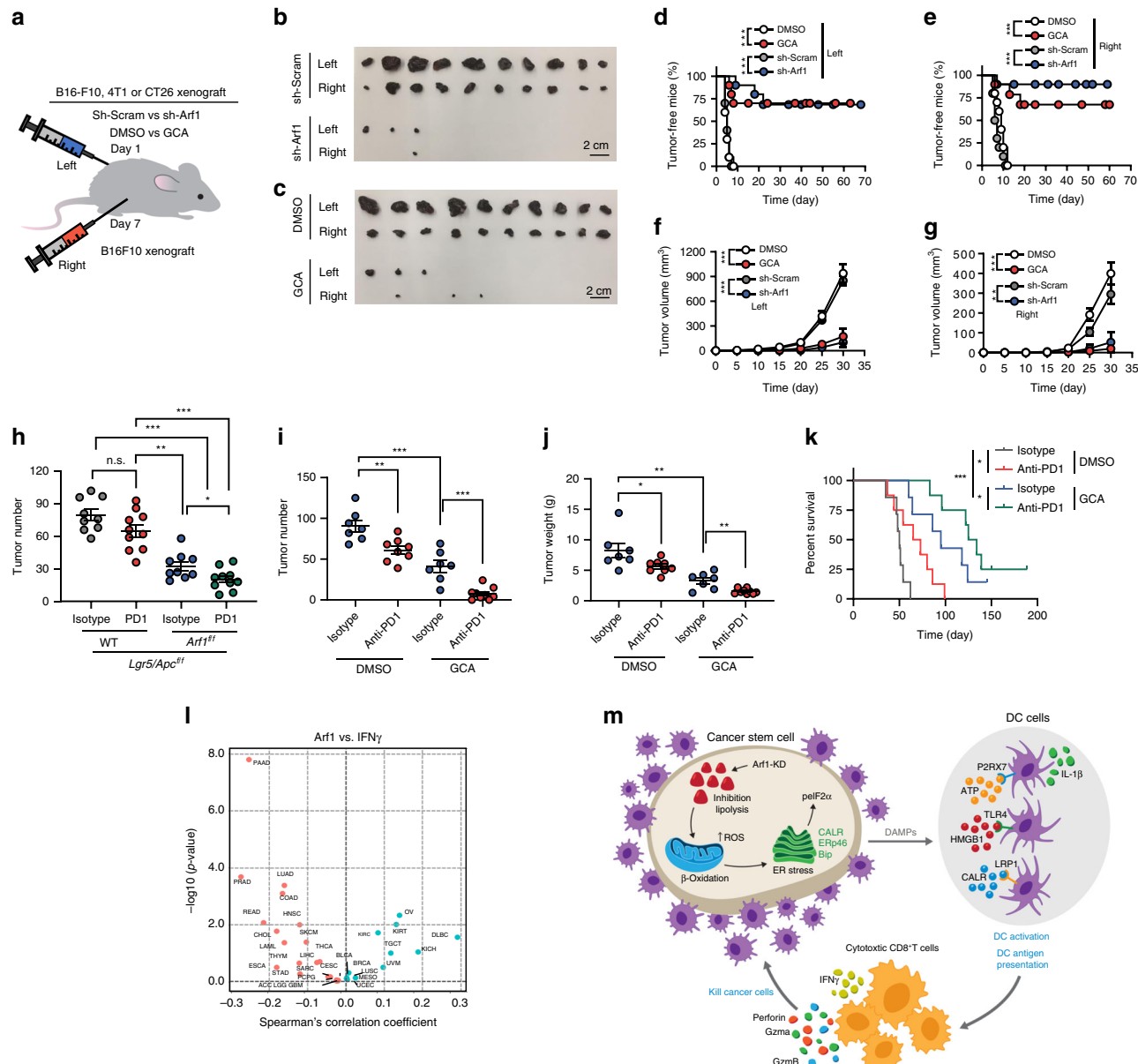

**Fig. 7 Vaccination with Arf1-ablated cells protects animals from developing tumors. a** Experimental setup. B16-F10 cells that were transfected with sh-Arf1 or sh-Scram or treated with the Arf1 inhibitor GCA or DMSO were injected into the left side of C57BL/6J mice, and the mice were then re-challenged 1 week later with untreated B16-F10 cells on the right side. **b, c** Tumor sizes from B16-F10 cells treated with the indicated reagents and transplanted into C57BL/6J mice. Scale bars: 2 cm. **d, e** Tumor-free curves of B16-F10 cells treated with the indicated reagents and transplanted into C57BL/6J mice. **f, g** Tumor-volume curves of B16-F10 cells treated with the indicated reagents and transplanted into C57BL/6J mice. **h** Intestinal tumor number of mice with the indicated genotypes after treatment with isotype or an anti-PD1 antibody. **i** Tumor numbers of MYC-ON mice treated with the indicated reagents. **j** Tumor weights of MYC-ON mice treated with the indicated reagents. **k** Survival curves of MYC-ON mice treated with the indicated reagents. **l** Correlation analysis for the Arf1 expression level versus IFNγ signature from a human cancer database. **m** Proposed model depicting how Arf1 knockdown promotes metabolic stress, the induction of DAMPs, and immune cell infiltration and activation to attack tumors (**b–g**: $n = 10$; **h**: $n = 10$; **i**: $n = 8$; **j**: isotype group, $n = 7$, anti-PD1 group, $n = 8$; **k**: $n = 8$ mice each group; $*p < 0.05$, $**p < 0.01$, $***p < 0.001$ by Student's $t$-test; polled two independent experiments). Data are shown as the mean ± SEM.

HMGB1, or BiP and treatment with oxATP, suramin, PBA, or GSK2606414 significantly blocked the effect of Arf1 inhibition, while the knockdown of ERp46 and treatment with ARL did not change the effect of Arf1 inhibition.

Collectively, these data suggest that the Arf1 ablation-induced anti-tumor immune responses occur through ER stress and DAMPs.

**Vaccination with Arf1-KO cells protects animals from developing tumors.** The immunogenicity of agents can be tested in a

mouse vaccination model[36,37]. In this model, we injected B16-F10 cells that were transfected with sh-Arf1 or sh-Scram or that were treated with the Arf1 inhibitor GCA or DMSO into the left side of C57BL/6J mice, and then rechallenged the mice 1 week later with untreated B16-F10 cells on the right side (Fig. 7a–g). The tumors in mice receiving B16-F10 cells treated with sh-Scram or DMSO grew progressively at both the left and right sides. Treatment of the injected cells with sh-Arf1 or GCA caused almost a complete regression of both the left- and right-side tumors. In situ

vaccination with Arf1 inhibition was effective not only against melanoma but also against tumors of a variety of histologic types, such as colon cancer (CT26) and breast carcinoma (4T1) (Supplementary Fig. 14a–c). These results demonstrated that Arf1 inhibition can trigger a T-cell immune response locally that then attacks cancer throughout the body.

Because the infiltrated T cells are frequently inhibited by immune inhibitory "checkpoint" receptors (in particular PD-1)[38], we further blocked PD-1 with anti-PD-1 mAbs in the Lgr5/Arf1/Apc intestinal CSC mice. We found that PD-1 blockage further reduced the tumor number (Isotype/Lgr5/Apc: 79.8 ± 15.4; PD1/Lgr5/Apc: 65.1 ± 18.2; Isotype/Lgr5/Arf1/Apc: 32.7 ± 12.3; PD1/Lgr5/Arf1/Apc: 20.4 ± 9.9) and increased the lifespan but did not affect the body weight of Lgr5/Arf1/Apc mice (Fig. 7h, and Supplementary Fig. 15a, b), indicating that Arf1 ablation and PD-1 blockage had a synergistic effect. We also tested Arf1 inhibition and PD-1 blockage in MYC-ON mice and found that PD-1 blockage could further reduce the tumor number (Isotype/MYC-ON: 90.6 ± 19.1; PD-1/MYC-ON: 61.0 ± 14.2; Isotype/MYC-ON/GCA: 41.1 ± 19.0; PD-1/MYC-ON/GCA: 7.7 ± 7.6) and tumor weight and increase the lifespan but did not affect the body weight of these mice (Fig. 7i–k, and Supplementary Fig. 15c, d). These data together suggest that Arf1 ablation and PD-1 blockage have a synergistic effect.

To determine the significance of our findings in human cancers, we analyzed public datasets on human cancer. We found that Arf1 was frequently amplified or overexpressed in the majority of cancer types examined (Supplementary Fig. 16a–c). In line with our finding that Arf1 inhibition increased the T-cell infiltration and activation in mouse models, the Arf1 expression level was found to be inversely correlated with IL-1β expression (Supplementary Fig. 16d), T-cell infiltration (Supplementary Fig. 16e), and INFγ expression (Fig. 7l) in a variety of cancer types in the The Cancer Genome Atlas (TCGA) cancer patient dataset. We also examined the relationship between the Arf1 expression level in tumors and clinical outcome. We found that the Arf1-low group had a significantly better survival probability than the Arf1-high group for a number of cancer types (Supplementary Fig. 17), suggesting that Arf1 is a negative prognostic factor for these tumors, with low Arf1 expression indicating a good prognostic outcome.

## Discussion

In this study, we identified a connection that links lipid metabolism, CSC death, and anti-tumor immunity. CSCs may be responsible for the treatment resistance and immune evasion of tumors. CSC signals can modulate lymphocyte infiltration into a tumor and alter the tumor microenvironment[2]. We demonstrated that ablating the Arf1-mediated lipid metabolism in CSCs resulted in metabolic stress and subsequent cellular responses, including the release of DAMPs, which promoted anti-tumor immunity by activating DCs, thereby enhancing T-cell infiltration and activation (Fig. 7m, and Supplementary Fig. 18). We further demonstrated that Arf1 ablation and PD-1 blockage had a synergistic effect. Consistent with these findings, TCGA data analysis showed an inverse correlation between Arf1 expression and T-cell infiltration and activation as well as better survival probability in various human cancers. Our results reveal that knockdown of the Arf1 pathway can have multimodal functions. It not only kills CSCs, but also elicits a tumor-specific immune response in which dying CSCs are converted into a therapeutic vaccine, resulting in durable efficacy of the treatment. To our best knowledge, this is the first demonstration that endogenous metabolic stress elicits anti-tumor immune responses by inducing DAMPs in rodent tumor models.

These pathogens express unique molecules, called pathogen-associated molecular patterns (PAMPs), which are recognized by pattern recognition receptors (PRRs) of the innate immune system to activate anti-infection immune responses. Such an induced immune response can be used to suppress tumors, as first demonstrated by William Coley[39]. In the 1890s, Coley successfully caused a number of sarcomas and/or lymphoma to regress after the intratumoral injection of streptococcal cultures in patients. It is assumed that some of the injected bacteria triggered PAMP-induced anti-tumor immune responses.

It was recently discovered that the immune system can not only be induced by PAMPs but also by endogenous signals called DAMPs, which are released from damaged or stressed cells in the absence of microbial components[17,19,28]. The DAMPs are "mimics of PAMPs". These molecules are often present in a given cell compartment and are not expressed or are only limitedly expressed under physiological conditions but are strongly induced and then are translocated to the cell surface or extracellular space under conditions of stress, damage, or injury. The DAMPs include Calr, ATP, HMGB1, double-stranded DNA (dsDNA), dsRNA, and CpG DNA. Under normal conditions, only microbial dsDNA or CpG DNA or viral dsRNA can be detected in the cytosol; however, under stressed or damaged conditions, cellular dsDNA, CpG DNA, and dsRNA may be released into cytosol. These DAMPs are recognized by PRRs: CpG DNA is recognized by TLR9, dsRNA by TLR3, MDA5, and RIG-1, and dsDNA by cytoplasmic DNA sensors, including the cyclic GMP-AMP synthase (cGAS). These interactions lead to the induction of type I interferons. Type I IFNs are critical mediators of the spontaneous priming of anti-tumor CD8[+] T-cell responses.

Interestingly, it was recently reported that low-dose DNA-demethylating agent (5-AZA-CdR) treatment[40] or histone demethylase LSD1 ablation[41] can induce the formation of dsRNA that "tricks" cancer cells into behaving like virus-infected cells, which promotes an MDA5/MAVS-dependent state of "viral mimicry" and stimulates anti-tumor immune responses. In the LSD1 situation, it was found that LSD1 inhibition induced both T-cell infiltration and PD-L1 expression, suggesting that LSD1 inhibition and checkpoint blockage may have synergistic anti-tumor effects[41]. In addition, some chemotherapy drugs induce the generation of dsDNA, which is recognized by cytosolic DNA sensors to elicit anti-DAMP (anti-virus-like) immune responses[19]. It was also reported recently that disruption of the LC3-associated phagocytosis (LAP) of the tumor-associated macrophages (TAMs) of dying tumor cells triggers STING-mediated type I interferon responses and anti-tumor immune responses, possibly through the release of undigested DNA from the engulfed tumor cells[42]. In the present study, we demonstrated that Arf1 inhibition resulted in lipid droplet accumulation, mitochondrial defects, ER stress, induction of DAMPs (Calr, ATP, HMGB1), and anti-tumor immune responses. Thus, the induction of anti-tumor immunity may prove to be another frontier of anti-tumor immune therapy, which can work alone to kill tumors or act in synergy with checkpoint blockers to achieve a potent effect.

We found that knocking down the Arf1-lipolysis pathway in CSCs triggered a chain reaction of metabolic stress, first causing mitochondrial damage and ER stress, then exposing DAMPs that recruit and activate DCs, which further activate IFN-γ-secreting cytotoxic T lymphocytes (CTLs) to kill CSCs, followed by tumor regression. Due to the well-known plasticity of tumor cells, only killing CSCs is not sufficient to elicit tumor regression[1]. In many patients, it was observed that a large number of antitumor T cells exist but remain in an inactive state due to an immunosuppressive environment. For the Arf1-knockdown-induced tumor regression, we propose the following sequence of events (Fig. 7m,

and Supplementary Fig. 18): a small number of active anti-CSC T cells penetrate the tumor and attack some CSCs; as a result of this interaction, these CTLs produce cytokines and re-stimulate the local environment around them to covert an immunosuppressive to an immunostimulatory environment; this not only reawakens the pre-existing inactive antitumor T cells in the tumors but also recruits new antitumor T cells; these additional T cells are directed against tumor antigens other than the DAMPs to amplify the effects for destroying the bulk tumors; these active T cells may then migrate to other tumor sites and become memory T cells to produce widespread and durable antitumor effects[43].

Our findings that Arf1 inhibition stimulates T-cell infiltration and activation may provide a basis for therapeutic strategies that exploit DAMP-mediated anti-tumor immunity induced in cancer patients. This may be another promising direction for cancer immunotherapies in addition to the successful checkpoint blockades. The therapies may include approaches that reduce Arf1 activity or block the Arf1-mediated lipolysis pathway and its associated β-oxidation.

## Methods

**Mice.** All the mice study protocol was approved by Animal Care and Use Committee (ACUC) at National Cancer Institute (NCI), NIH. The relevant ethical regulations for animal testing and research followed the regulation of ACUC at NCI and approved by ACUC at NCI, NIH. Arf1$^{f/f}$ mice were generated on the C57BL/6 background using the FloxP method, at Dr. Lino Tessarollo's mouse facility in the Mouse Cancer Genetics Program (MCGP) at NCI-Frederick, NIH. The mouse Arf1 embryonic stem cells were purchased from the International Mouse Phenotyping Consortium (IMPC). Foxa3-Cre mice were purchased from Mutant Mouse Resources & Research Centers (MMRRC) (Cat#011121-MU)[44]. The other mice were generated by other laboratories as follows: Albumin-Cre (Mark A. Magnuson, Vanderbilt University), Lgr5-EGFP-IRES-creERT2 (Hans Clevers, Hubrecht Institute), Apc$^{f/f}$ (Tyler Jacks, Massachusetts Institute of Technology), Cebpb-tTA (IMR Colony, The Jackson Laboratory), tet-o-MYC (J. Michael Bishop, University of California, San Francisco), Axin2-CreER (Roel Nusse, Stanford University), Rag1-KO (IMR Colony, The Jackson Laboratory), IFNg-KO (Timothy Stewart, Genentech Inc., South San Francisco), NU/J, R26R (IMR Colony, The Jackson Laboratory). All the mice except for Foxa3-Cre were purchased from The Jackson Laboratories and crossed to Arf1$^{f/f}$ mice.

Lgr5/Arf1$^{f/f}$/Apc$^{f/f}$, Lgr5/Arf1$^{f/f}$/Apc$^{f/f}$/Rag1-KO, Lgr5/Arf1$^{f/f}$/Apc$^{f/f}$/IFNg-KO, Axin2-CreER/R26R/Arf1$^{f/f}$, Cebpb-tTA/tet-o-MYC, Cebpb-tTA/tet-o-MYC/Axin2-CreER/Arf1$^{f/f}$ mice were generated by crossing the different lines with Arf1$^{f/f}$ mice. For the FloxP and KO mice, heterozygous or wild-type (WT) mice were used as the control. All mice were housed with chow and water supplied automatically, under pathogen-free conditions and a 12-h dark/light cycle. The mice were approved to use the animal holding space from the CCR Animal Resource Program Office, all procedures were performed in accordance with the NCI Animal Care and Use Committee (ACUC) at NIH, and the facility person fully followed the instructions for the animal facility operating procedures. Age- and sex-matched colonies or strains of male and female mice were used in the in vivo KO and subcutaneous tumor xenograft experiments, and the appropriate littermate controls were maintained under the same conditions.

**Cells.** The CT26 colorectal carcinoma cell line (derived from BALB/c mice) (ATCC, Cat#CRL-2638™), B16-F10 skin melanoma cell line (derived from C57BL/6J mice) (ATCC, Cat#CRL-6475™), A20 B cell lymphoma cell line (derived from BALB/cAnN mice) (ATCC, Cat#TIB-208™), and 4T1, an animal stage IV breast cancer cell line (derived from BALB/cfC3H mice) (ATCC, Cat#CRL-2539™) were purchased from ATCC. The Huh-7 human liver cell line was obtained from Creative Biolabs. The 293T human embryonic kidney cell line was purchased from ATCC (ATCC, Cat#CRL-11268™). The CT26, A20, 4T1, and Huh-7 cell lines were maintained in RPMI1640 medium containing 10% FBS, 100 units/mL penicillin/streptomycin, 2 mM L-glutamine, 50 μg/mL Normocin, and 2.0 g/L sodium bicarbonate. The B16-F10 and 293T cells were cultured in Dulbecco's Modified Eagle Medium (DMEM) containing 10% fetal bovine serum (FBS), 100 units/mL penicillin/streptomycin, 4 mM L-glutamine, 4.5 g/L glucose, 1 mM sodium pyruvate, and 1.5 g/L sodium bicarbonate. All cell lines were cultivated in a humidified atmosphere containing 5% $CO_2$ at 37 °C. The use of the human cell line for xenografts in mice was approved by the Animal Care and Use Committees (ACUC) and registered with the Institutional Biosafety Committee (IBC) at NCI-Frederick, NIH.

**In vivo induced mouse models.** To induce colon tumors in mice, the Lgr5/Arf1$^{f/f}$/Apc$^{f/f}$ mice were intraperitoneally (i.p.) injected with 10 mg/kg tamoxifen (Sigma-

Aldrich, Cat#T5648) dissolved in corn oil (Sigma-Aldrich, Cat#C8267) each day for 3 consecutive days. To induce liver and colon tumors in Axin2-CreER/R26R/Arf1$^{f/f}$ and Axin2-CreER/R26R/Arf1$^{f/f}$/Apc$^{f/f}$ mice, 10 mg/kg tamoxifen dissolved in corn oil was i.p. injected each day for 5 consecutive days.

For the liver tumor model in mice, Cebpb-tTA/tet-o-MYC and Cebpb-tTA/tet-o-MYC/Axin2-CreER/Arf1$^{f/f}$ mice, the food in the mating cage and the wean cages was a grain-based rodent diet containing 200 mg/kg doxycycline (Bio-Serv, Cat#14-727-450). Tumors were induced in 6-week-old mice, and the chow was changed to a normal diet. In the Cebpb-tTA/tet-o-MYC/Axin2-CreER/Arf1$^{f/f}$ mice, the Arf1 gene was deleted in the stem cells by the i.p. injection of 10 mg/kg tamoxifen (Sigma-Aldrich, Cat#T5648) each day for 5 consecutive days. The mice were euthanized after 8–10 weeks.

**Engrafted tumor models.** For grafts of mouse tumor cells in BALB/c or C57BL/6J background mice, littermate mice were separated into groups, and $5 \times 10^5$ CT26 cells, $5 \times 10^6$ A20 cells, or $1 \times 10^4$ 4T1 cells were injected subcutaneously (s.c.) into the mouse's left or right side. The tumor volume was measured by a digital caliper (ULINE, Cat#H-7352), and the tumor volume was calculated using the formula: ½ × longitudinal diameter (length) × the greatest transverse diameter (width)[2]. Mice were euthanized when the tumor volume reached 2000 mm³, or 70 days after the engraftment of tumor cells.

**Anti-tumor vaccination experiments.** A20, CT26, 4T1, and B16-F10 tumor cells treated with 2.5 μM Arf1 inhibitor GCA or with Arf1 RNAi ($5 \times 10^6$, $5 \times 10^5$, $1 \times 10^4$, and $5 \times 10^4$ cells, respectively) were s.c. inoculated into the left side of the abdomen of 6-week-old female mice. Seven days later, the same amount of corresponding untreated tumor cells was inoculated into the right side. The tumor cell lines were mixed with 50% growth-factor-reduced Matrigel (Gibco, Cat#354234) or resuspended in phosphate-buffered saline (PBS) and injected s.c. or intradermally.

**APAP-induced liver injury.** APAP was purchased from Sigma-Aldrich and dissolved in PBS at 20 mg/kg freshly for each experiment. APAP was used at 300 mg/kg and fed by gavage needle after mice were starved for 12 h. Blood was collected from mice by eye removal, and then it was centrifuged by 13000 g for 5 minutes to separate the serum. Liver was isolated from the mice for immunohistochemistry analysis. Mice were euthanized by $CO_2$ for 30 min if they became moribund.

**Generation of cell lines.** The shRNA oligonucleotides were synthesized by Macrogen Ltd. (USA); the detailed sequences are listed in Table S1. The double oligonucleotides were annealed at 95 °C for 20 min and cloned into the pLKO.1-Puro lentivirus vector. After sequencing the pLKO.1 plasmid containing different shRNAs, lentiviruses carrying the pLKO.1 plasmid were produced by co-transfecting HEK293T cells with three helper plasmids: pRSV-Rev, pMD2.G, and pMDLg/pRRE. The lentiviral particles were harvested from the medium after 48 h of co-transfection. The cell medium was passed through a 0.22-μm filter, and the lentivirus was collected into Eppendorf tubes and kept at −20 °C for short-term or −80 °C for long-term storage. The cells were infected and selected by 5 μg/mL puromycin (Thermo Fisher, Cat#A1113803) or 0.8 mg/mL G418 (Sigma-Aldrich, G8168) for 20 days. Stable cell lines were cultured in 1.0 μg/mL puromycin or 0.4 mg/mL G418. Cells were transfected with the Arf1 siRNA using the Lipofectamine™ RNAiMAX (Thermo Fisher, Cat#13778075) reagent, harvested 48 h later, and engrafted into mice.

**T-cell depletion.** Mice were treated with 100 μg/mouse of rat anti-CD4 (clone GK1.5, BioXcell, Cat#BE0003-1), 100 μg/mouse of rat anti-CD8α (clone 2.43, BioXcell, Cat#BE0061), or 100 μg/mouse of rat IgG (BioXcell, Cat#BE0094) isotype control antibodies. All antibodies were diluted in PBS and i.p. injected into mice. For Lgr5-CreER/Arf1$^{f/f}$/Apc$^{f/f}$ and control mice, antibodies were injected on each of 3 consecutive days, followed by the injection of tamoxifen on each of 3 consecutive days, and then the antibodies were injected on each of 3 consecutive days. For the MYC-ON liver tumor model, the antibodies were injected once a week, for 8–10 consecutive weeks, with the first injection performed after the doxycycline chow had been withdrawn for 1 week.

**MYC-ON mice treated with GCA, BFA and T-cells deletion.** For MYC-ON liver tumor model, the antibodies were i.p. injected per week, and continued injection for 5 weeks, the first injection after doxycycline chow had been withdrawn for 1 week. The antibodies used for treatment mice as follows: 100 μg/mouse of rat anti-CD4 (clone GK1.5, BioXcell, Cat#BE0003-1), 100 μg/mouse of rat anti-CD8α (clone 2.43, BioXcell, Cat#BE0061) or 100 μg/mouse rat IgG (BioXcell, Cat#BE0094) Isotype controls antibodies; 100 μg/mouse of Armenian Hamster anti-mouse PD1 (J43, BioXcell, Cat# BE0033-2) and 100 μg/mouse of Armenian Hamster Isotype IgG (BioXcell, Cat#BP0091) were used as control. GCA (2 mg/kg), BFA (4.0 mg/kg), and Tunicamycin (2.0 mg/kg) dissolved in corn oil were administrated by gavage needle for continuously 4 weeks from the second week of MYC-ON; the mice were monitored every 2 days, and after 10 weeks the mice were euthanized and the tumor growth was analyzed. For survival days, after deletion of

the T cells with antibodies, the mice were treated with GCA (2.0 mg/kg) or DMSO for continuously 4 weeks and monitored every 2 days until the mice died.

**Isolation of immune cells**. To isolate spleen cells, a mouse was euthanized by $CO_2$ for 20 min, and the spleen was harvested, minced into small pieces, and placed in RPMI 1640 medium (Quality Biological, Cat#112-025-101). The minced spleen was dissociated by pipetting up and down with a 5-mL syringe plunger (The Lab Depot, Inc., Cat#BD309647). The cells were then passed through a 70-μm strainer (Celltreat, Cat#229484), followed by the lysis of red blood cells. The isolated cells were washed twice in cell staining buffer (0.5% BSA in 1× PBS), and subjected to gradient centrifugation in 33% percoll (Sigma-Aldrich, Cat# P1644). The pellet was washed twice with cell staining buffer and stained for flow cytometry analysis.

Mouse intestine IEL and LPL immune cells were isolated as follows. The colon and small intestine were removed from a mouse and cleared of all connective tissue and fat, and the small intestine (SI)/colon was cleared of fecal matter. All the peyers patches were cut off small intestine after clearing the fecal matter. The small intestine/colon was then washed in ice-cold RPMI 1640 medium three times for 5 min each. The SI/colon was cut into 1–3-cm pieces in length and placed in a 50-mL beaker with 10 mL of pre-warmed IEL isolation medium (RPMI 1640, 100 units/mL penicillin/streptomycin, 3% FBS, 5 mM EDTA, and 0.145 mg/mL DTT). The SI/colon pieces were then placed into a flask with a stir bar, covered with aluminum foil, and incubated for 20 min at 37 °C with stirring. The contents of the flask were then passed through a sterile 200-μm kitchen strainer into a 100 mL beaker on ice. The pieces of the SI/colon were then transferred into a 50-mL tube, and 10 mL of shaking medium (RMPI 1640, 100 units/mL penicillin/streptomycin, 2 mM EDTA) was added. The tube was vigorously shaken for 30 s, and the contents were then passed through a 200-μm kitchen strainer into the same beaker. The shaking and straining steps were then repeated two more times. The beaker on ice contained the IELs.

To isolate IELs, the solution in the beaker was passed through a 70-μm cell strainer into a 50-mL tube, and then spun down at $350 \times g$ for 7 min at 4 °C. The pellet was resuspended in RPMI 1640 + 3% FBS and passed through a 40-μm cell strainer (Celltreat, Cat#229482). The cells were spun down again, the pellet was resuspended in 40 mL of 30% percoll (at room temperature), and the cells were centrifuged for 20 min at $400 \times g$. After centrifugation, the supernatant was carefully removed using a pipette. The cell pellet was then resuspended in RPMI 1640 + 3% FBS and transferred to a clean tube. The cell pellet was washed at least once before staining.

To isolate LPLs from the SI/colon, the pieces of SI/colon were placed into a 50-mL beaker containing 10 mL of RMPI 1640, 100 units/mL penicillin/streptomycin, Liberase (I:250), and 0.05% DNase (NO SERUM). The gut pieces were minced with scissors, then a stir bar was added, and the pieces were incubated for 30 min at 37 °C with stirring. After incubation, the mixture was poured through a 70-μm cell strainer into a 50-mL tube. The remaining pieces of the gut were mashed through the strainer. The beaker was washed with RMPI 1640 + 3% BSA and the wash also passed through the strainer. The cell suspension was centrifuged for 7 min at $350 \times g$. The pellet was resuspended in RMPI 1640 containing 3% BSA and passed through a 40-μm filter. Then cells were spun down, and the cell pellet was resuspended in 30% percoll in 1× PBS. The contents were mixed well and then centrifuged for 10 min at $350 \times g$ at 4 °C. This pellet was the LPL immune cells.

Mouse liver immune cells were isolated by dissecting the entire liver and placing it into a 60-mm Petri dish. The liver was minced into small pieces and homogenized with a syringe plunger against a 70-μm mesh screen. Liver cells were collected and passed through a 70-μm strainer into a 50-mL tube. The single-cell suspension was placed into 33% percoll, mixed well, and centrifuged for 20 min at $300 \times g$ at room temperature. This pellet contained red blood cells and lymphocytes. The pellet was washed with RBC lysis buffer (Biolegend, Cat#420301) for 3 min, and the cells were spun for 5 min at $350 \times g$ at 4 °C and resuspended in RPMI medium. The cells were then ready to stain for flow cytometry.

**Flow cytometry analysis of immune cells**. For flow cytometry, single-cell suspensions in cell staining buffer (0.5% BSA in 1× PBS) were subjected to red blood cell lysis buffer (Biolegend, Cat#420301) for 3 min, and the cells were then blocked by TruStain fcXTM (anti-mouse CD16/CD32) (Biolegend, Cat#101319) to reduce nonspecific immunofluorescent staining, at room temperature for 10 min. The living and dead cells were discriminated by the LIVE/DEAD™ Fixable Aqua Dead Cell Stain Kit (Thermofisher, Cat#L34965) according to the manufacturer's instructions. Cell surface staining was performed by adding appropriately conjugated fluorescent primary antibodies to the cells and incubating on ice for 30 min in the dark. After staining, the cells were washed three times for 5 min each with cell staining buffer and fixed with 4% paraformaldehyde (PFA) for flow cytometry analysis. For intracellular staining, the cells after surface staining were permeabilized by Intracellular Staining Permeabilization Wash Buffer (Biolegend, Cat#421002), and the intracellular staining was carried out according to the manufacturer's instructions. For nuclear staining, the cells were fixed and permeabilized using the True-Nuclear™ Transcription Factor Buffer Set (Biolegend, Cat#424401), and then the nuclear proteins were stained by adding the appropriate fluorescent primary antibodies to the cells and incubating on ice for 30 min in the dark. After the intracellular or nuclear staining, the cells were washed with cell staining buffer twice for 5 min each at room temperature and analyzed by flow

cytometry. The antibodies used are listed in the Key Resource Table. The detail gating strategy for sorting the immune cells is shown in the Supplementary Figs. 19 and 20.

**Plasmids**. The pRSV-Rev, pMD2.G, pMDLg/pRRE, pLKO.1-puro, and pLKO.1-puro-GFP were purchased from Addgene. The pmeLUC was kindly provided by Dr. Di Virgilio Francesco of the University of Ferrara. The human and mouse Arf1 shRNAs, and human and mouse Calr shRNAs constructed in pKO.1-puro plasmids, were purchased from Sigma-Aldrich. The other plasmids were generated in lab; the detailed sequences are shown in Supplemental Table 1. Briefly, all the shRNAs were synthesized by Macrgene, and the pLKO.1-Puro plasmids were double digested by the AgeI-HF (New England Biolabs, Cat#R3552S) and EcoRI-HF (New England Biolabs, Cat#R3101S) enzymes. The shRNAs and vector were ligated by T4 DNA ligase (New England Biolabs, Cat#M0202S). Positive clones were sequenced by Macrogene to verify that the ligation was correct.

**Immunohistochemistry and immunofluorescence**. For the immunohistochemistry (IHC) and IF of tissues, mouse tissues or graft tumors were dissected after the mice were euthanized by $CO_2$ for 20 min and fixed in 10% neutral buffered formalin solution (10% NBF) (Sigma-Aldrich, Cat# HT501128) at 4 °C overnight with shaking. The next day, the tissues were embedded in paraffin wax or OCT (Tissue-Tek; Sakura Finetek USA, Cat#4583) and sectioned with a Leica RM2125 Microtome for paraffin sections or a Leica CM1950 for frozen sections. The paraffin-embedded sections were deparaffinized with xylene and alcohol and subjected to antigen retrieval by citrate buffer (Abcam, Cat# ab93678), Tris-EDTA buffer (Abcam, Cat# ab93684), EDTA buffer (Abcam, Cat# ab93680), or Tris buffer (Abcam, Cat# ab93682) in a 100 °C cooker for 30 min. The cryosections and antigen retrieval paraffin sections were washed in 1× PBST (PBS-Tween-20) three times for 5 min each, and blocked in 10% goat serum, 1% BSA for 1 h. After blocking, the sections were incubated with primary antibodies at 4 °C overnight. The next day, the slides were washed with 1× PBST three times for 5 min each and stained with the secondary antibodies and Hoechst 60 min. For IHC staining, the mouse tissue sections were developed with an IHC Detection kit (Thermo Fisher, Cat#3600) according to the manufacturer's instructions. For cell IF, the cells were fixed in 4% PFA (Thermo Fisher, Cat#28906) for 30 min, and washed with 1× PBST three times for 5 min each, and then the cells were incubated with blocking buffer (10% goat serum, 1% BSA) for 1 h. The primary antibodies were diluted in 1% BSA, and the cells were incubated with primary antibodies at 4 °C overnight. The next day, the slides were washed with 1× PBST three times for 5 min each and stained with fluorescently labeled secondary antibodies for 1 h. The tissue sections or cells were examined with a Zeiss (LSM780) confocal microscope system, and figures were analyzed using Zeiss blue software.

**Antibodies**. For IHC, IF, and WB:
  IHC & IF: 1:200, Mouse anti-β-Catenin (BDI080), Abcam, Cat#Ab19448
  IHC & IF: 1:500, Chicken anti-GFP, Abcam, Cat#Ab13970
  IHC & IF: 1:200, Rabbit anti-CD3ε (D4V8L), Cell Signaling, Cat#99940
  IHC & IF: 1:200, Rabbit anti-CD4 (EPR19514), Abcam, Cat# ab183685
  IHC & IF: 1:150, Rat anti-CD8a Antibody (4SM15), eBioscience, Cat#14-0808-82
  IHC & IF: 1:150, Rabbit anti-pZap-70 (65E4), Cell Signaling, Cat#2717
  IHC & IF: 1:100, Mouse anti-mouse I-AK (AαK) (MHC-II) (11-5.2), Biolegend, Cat#110002
  IHC & IF: 1:100, Mouse anti-mouse H-2Kb/H-2Db, Biolegend, Cat#114602
  IHC & IF: 1:300, Rabbit anti-Calreticulin Antibody, Thermo Fisher, Cat#PA3-900
  IHC & IF: 1:200, Mouse anti-ERp46 Antibody (C-11), Santa Cruz, Cat#sc-271667
  IHC & IF: 1:100, Rabbit anti-HMGB1 Polyclonal Antibody, Thermo Fisher, Cat#PA1-16926
  IHC & IF: 1:100, Mouse anti-human LAMP-1 Antibody (H4A3), Biolegend, Cat#328601
  IHC & IF: 1:150, Rabbit anti-eIF2A (pSer51) Antibody, Novus Biologicals, Cat#NB100-81896
  IHC & IF: 1:100, Rabbit anti-Phospho-eIF2α (Ser51), Cell Signaling, Cat#9721
  IHC & IF: 1:100, WB: 1:1000, Rabbit anti-Cleaved Caspase-3 (Asp175) (5A1E), Cell Signaling, Cat#9664S
  IHC & IF: 1:150, Mouse anti-Caspase-1(p20), AdipoGen Life Science, Cat#AG-20B-0042-C100
  IHC & IF: 1:200, Rabbit anti-PEAR1 Polyclonal Antibody, Thermo Fisher, Cat#PA5-21057
  IHC & IF: 1:150, Armenian Hamster anti-CD11c Antibody, Novus Biologicals, Cat#NB110-97871
  IHC & IF: 1:100, Mouse anti-LRP1 Monoclonal Antibody, Thermo Fisher, Cat#37-3800
  IHC & IF: 1:200, Mouse anti-KDEL (10C3), Novus Biologicals, Cat#NBP1-97469
  IHC & IF: 1:150, WB: 1:2000, Mouse anti-HNF-4-alpha antibody [K9218], Abcam, Cat#ab41898

IHC & IF: 1: 250, WB: 1:2000, Rabbit anti-alpha 1 Fetoprotein antibody, Abcam, Cat#ab46799

IHC & IF: 1:150, WB: 1:1000, Rabbit anti-Albumin antibody, Abcam, Cat#ab207327

WB: 1:1000, Mouse anti-HNF-6 Antibody (G-10), Santa Cruz, Cat#sc-376167

WB: 1:1000, Mouse anti-Cytokeratin 19 Antibody (A-3), Santa Cruz, Cat#sc-376126

WB: 1:2000, Mouse anti-Hex antibody, Abcam, Cat#Ab117864

IHC & IF: 1:200, WB: 1:2000, Rabbit anti-ARF1 Polyclonal Antibody, Thermo Fisher, Cat#PA1-127

WB: 1:1000, Mouse anti-GAPDH Antibody (GA1R), Thermo Fisher, Cat#MA5-15738

IHC & IF: 1:150, WB: 1:2000, Mouse Anti-E Cadherin antibody (M168), Abcam, Cat#Ab76055

WB: 1:1000, Rabbit anti-CEBP Alpha/CEBPA antibody, Abcam, Cat#Ab40764

WB: 1:2000, Rabbit anti-Cytokeratin 17 antibody, Abcam, Cat#Ab53707

WB: 1:1500, Rabbit anti-Cytokeratin 7 antibody, Abcam, Cat#Ab181598

WB: 1:500, Mouse anti-GADD 153 antibody (B-3), Santa Cruz, Cat#sc-7351

IHC & IF: 1:100, WB: 1:2000, Rabbit anti-GRP78/HSPA5 Antibody, Novus Biologicals, Cat#NB300-520

IHC & IF: 1:150, Goat anti-Mouse IL-1 beta /IL-1F2 Antibody, R&D Systems, Cat#AF-401-NA

WB: 1:2000, Rabbit anti-Cyclin D1 antibody (EPR2241), Abcam, Cat#ab134175

IHC & IF: 1: 300, WB: 1:2500, Mouse anti-SQSTM1/p62 antibody, Abcam, Cat#Ab56416

WB: 1:500, Mouse anti-Phospho-p70 S6 Kinase (Thr389) (1A5), Cell Signaling, Cat#9206S

For flow cytometry, all of the FACS antibodies were used as 1:100 dilution by FACS buffer.

TruStain FcX™ (anti-mouse CD16/32) Antibody, Biolegend, Cat#101320

APC/Cy7 anti-mouse CD45 Antibody, Biolegend, Cat#103116

Pacific Blue™ anti-mouse CD3 Antibody, Biolegend, Cat#100213

Alexa Fluor® 647 anti-mouse CD8a Antibody, Biolegend, Cat#100727

PerCP/Cyanine5.5 anti-mouse CD4 Antibody, Biolegend, Cat#116011

FITC anti-mouse TCR β chain Antibody, Biolegend, Cat#109205

PE anti-mouse TCR γ/δ Antibody, Biolegend, Cat#118107

BV605™ anti-mouse NK-1.1 Antibody, Biolegend, Cat#108740

BV785™ anti-mouse CD25 Antibody, Biolegend, Cat#102051

BV421™ anti-mouse/human CD11b Antibody, Biolegend, Cat#101251

FITC anti-mouse Ly-6G/Ly-6C (Gr-1) Antibody, Biolegend, Cat#108405

BV650™ anti-mouse F4/80 Antibody, Biolegend, Cat#123149

BV785™ anti-mouse CD11c Antibody, Biolegend, Cat#117336

PE anti-mouse/human CD45R/B220 Antibody, Biolegend, Cat#103208

PerCP/Cyanine5.5 anti-mouse I-A/I-E Antibody, Biolegend, Cat#107625

PE/Cy7 anti-mouse CD127 (IL-7Rα) Antibody, Biolegend, Cat#135013

Human/Mouse/Rat FoxP3 AF 405-conjugated Antibody, R&D Systems, Cat#IC8970V

PE anti-mouse CD8a Antibody, Biolegend, Cat#100707

Alexa Fluor® 488 anti-mouse CD4 Antibody, Biolegend, Cat#100425

PE Hamster Anti-Mouse γδ T-Cell Receptor, BD Biosciences, Cat# 561997

Mouse IFN-gamma R2 Alexa Fluor® 405-conjugated Antibody, R&D Systems, Cat#FAB773V

TNF alpha Rat anti-Mouse, PE-eFluor 610, Life Technologies, Cat#LS61732182

PE/Cy7 anti-mouse IL-17A Antibody, Biolegend, Cat#506921

APC anti-mouse CD8a Antibody, Biolegend, Cat#100711

Alexa Fluor® 700 anti-mouse CD45 Antibody, Biolegend, Cat#103127

APC/Cy7 anti-mouse TCR β chain Antibody, Biolegend, Cat#109219

PE anti-mouse IL-17A Antibody, Biolegend, Cat#506903

PE/Cy7 anti-mouse NK-1.1 Antibody, Biolegend, Cat#108713

BV510™ anti-mouse CD4 Antibody, Biolegend, Cat#100449

PE anti-mouse CD8a Antibody, Biolegend, Cat#100707

APC anti-mouse IL-17A Antibody, Biolegend, Cat#506915

APC/Cy7 anti-mouse Ly-6G Antibody, Biolegend, Cat#127623

APC anti-mouse Ly-6C Antibody, Biolegend, Cat#128015

PE/Cy7 anti-mouse CD19 Antibody, Biolegend, Cat#115519

PerCP/Cy5.5 anti-mouse/human CD45R/B220 Antibody, Biolegend, Cat#103235

PE-eFluor 610 F4/80 Monoclonal Antibody (BM8), ThermoFisher, Cat#61-4801-82

PE anti-mouse I-A/I-E Antibody, Biolegend, Cat#107607

PerCP anti-mouse CD11c Antibody, Biolegend, Cat#117325

AF 405 anti-mouse CD11b Antibody (M1/70), Novus Biologicals, Cat#FAB1124V-025

APC anti-mouse CD8b Antibody, Biolegend, Cat#126613

PerCP anti-mouse CD25 Antibody, Biolegend, Cat#102027

BV510™ anti-mouse CD4 Antibody, Biolegend, Cat#100449

PE-eFluor 610 Ki-67 Monoclonal Antibody, Thermo Fisher, Cat#61-5698-82

PE/Cy7 anti-mouse CD105 Antibody, Biolegend, Cat#120409

APC anti-GATA3 Antibody, Biolegend, Cat#653805

BV786 Rat Anti-Mouse IL-17A, BD Biosciences, Cat#564171

BV605 Rat Anti-Mouse CD119 (IFN-γ), Biolegend, Cat#745111

IL-13 Monoclonal Antibody (eBio13A), PE-Cyanine7, Thermo Fisher, Cat#25-7133-82

BV711 Rat Anti-Mouse TNFa, BD Biosciences, Cat#563944

Brilliant Violet 421™ anti-human TCR Vα7.2 Antibody, Biolegend, Cat#351715

BV711 Rat Anti-Mouse CD1d, BD Biosciences, Cat#740711

FITC-CD44 anti-human mouse antibody, Biolegend, Cat# 103005

PE/Cy7-CD133 anti-mouse atibody, Biolegend, Cat#141210

PE/Cy7-CD133 anti-human atibody, Biolegend, Cat#372810.

**Western blotting and electron microscopy.** Cell or mouse tissue proteins were released by RIPA lysis buffer (0.1% SDS, 150 mM NaCl, 0.5% sodium deoxycholate, 1 mM EDTA, 1 mM EGTA, 1.0% Triton X-100, and 50 mM Tris pH 8.0) containing 1× protease inhibitor cocktail (Promega) and phosphatase inhibitor cocktail (Sigma-Aldrich). Mouse tissue was homogenized in RIPA buffer for 30 times and lysed on ice for 30 min. For cells, the cells were collected from the culture dish, lysed by RIPA buffer for 30 min on ice. The tissues or cells were then centrifuged at $12,000 \times g$ for 3 min at 4 °C. The supernatant protein concentration in the lysates was determined by a Bio-Rad Protein Assay kit (Bio-Rad, Cat#5000006). All samples were adjusted to 3.0 μg/μL protein, 5× SDS loading buffer was added, and the sample was boiled at 100 °C for 15 min. Equal volumes of protein samples were loaded onto 4–20% Mini-PROTEAN® TGX™ Precast Protein Gels (Bio-Rad, Cat#4561096), separated by electrophoresis with a 120 V PowerPac™ Basic Power Supply (Bio-Rad) for 65 min, and transferred to a nitrocellulose membrane (GE Healthcare, Cat#10600014). The membrane was blocked with 5.0% Blotting-Grade Blocker (Bio-Rad, Cat#1706404) for 2 h, and then incubated with the appropriate primary antibody at 4 °C overnight with shaking. The next day, the membrane was washed with 1× PBST three times for 5 min and incubated with the secondary HRP antibody (1:2000) in 5% milk for 2 h. The membrane was then developed by Pierce™ ECL Western Blotting Substrate (Thermo Fisher, Cat#32106), applied to HyBlot CL® Autoradiography film (Thomas Scientific, Cat#3022), and developed with an S&W Imaging system (Quantum Medical Enterprises LNC.). The western blot uncropped images are shown in the Supplementary Fig. 21.

For EM, pregnant mice were euthanized with $CO_2$ for 30 min, an embryo was isolated from the maternal uterus, and the embryo liver was dissected. The intestine was isolated from Lgr5-CreER, Lgr5-CreER/Arf1[f/f], Lgr5-CreER/Apc[f/f], or Lgr5/Arf1[f/f]/Apc[f/f] mice. The liver or intestine was perfused with EM buffer (4% PFA, 0.2% glutaraldehyde, 0.1 M sodium cacodylate) for 2 days. The EM was performed by the Electron Microscopy Core in the Advanced Technology Research Facility of the Center for Cancer Research (CCR) NCI-Frederick, NIH.

**X-gal staining.** Mouse tissues were collected, perfused with 4% PFA, and incubated on an end-to-end shaker at 4 °C overnight. The next day, the tissues were embedded in paraffin or OCT (Tissue-Tek; Sakura Finetek USA, Cat#4583) and sectioned with a Leica RM2125 Microtome for paraffin sections or a Leica CM1950 for frozen sections. The tissues were rinsed with rinse buffer: 0.1 M sodium phosphate dibasic (Sigma-Aldrich, Cat#S7907), 5.0 mM sodium phosphate monobasic (Sigma-Aldrich, Cat#S9638), 3.0 mM magnesium chloride hexahydrate (Sigma-Aldrich, Cat#M2670), 1.5 mM sodium deoxycholate (Sigma-Aldrich, Cat#30970), and 3.0% IGEPAL CA-630 (Sigma-Aldrich, Cat#I8896) for 30 min at 4 °C, and washed with rinse buffer two more times at room temperature for 30 min each. The tissues were then stained with staining buffer: 5.0 mM potassium ferricyanide (Sigma-Aldrich, Cat#702587), 5.0 mM potassium ferrocyanide (Sigma-Aldrich, Cat#P3289), and 1.0 mg/mL X-gal (Sigma-Aldrich, Cat# B4252) at 37 °C overnight. After staining overnight, wash the tissue with 70% ethanol for two times, each times 30 min, and then the tissues fixed in 4% PFA[45].

**RNA extraction and qRT-PCR.** For quantitative reverse transcription polymerase chain reaction, the total RNA was extracted from the mouse intestine, liver, or xenograft tumors with the Monarch® Total RNA Miniprep Kit (New England Biolabs, Cat#T2010S) or RNeasy Micro Kit (Qiagen, Cat#74104), and the cDNA was synthesized by the High-Capacity cDNA Reverse Transcription Kit (Applied Biosystems, Cat#4368814). The primers used for qRT-PCR are listed in Table S1. The total cDNA was quantified using a NanoDrop (DS-11 Spectrophotometer from DeNovix Inc.), and amplification was performed by the SYBR™ Select Master Mix (Applied Biosystems, Cat#4472903) with a CFX96 Touch™ Real-Time PCR Detection System (Bio-Rad). The Gapdh gene was used as an internal control. The results were calculated as $2^{-\Delta CT}$.

**B16-F10 tumor metastasis.** C57BL/6J mice were used to study the tumor metastasis, the mice were anesthetized by Isothesia (HENRY SCHEIN, Cat#NDC11695-6776-2), and $5 \times 10^4$ B16-F10 cells stable transfected with sh-Scramble or sh-Arf1 were injected into tail vein of each mouse. After 15 days, the mice were euthanized with $CO_2$ for 30 min and the metastasis tumors in different organs were analyzed .

**In vivo photograph of ATP release.** CT26 cells were transfected with pmeLUC plasmid as shown previously[46]. After transfection for 48 h, the cells were selected with G418 (1.0 mg/mL) (Sigma-Aldrich, Cat#G8168) for 30 days. The stable cell line was kept in 0.5 mg/mL G418. BALB/c mice were transplanted with $5 \times 10^5$

CT26 cells that stable expressed pmeLUC, the tumor was monitored every 5 days and calculated with digital caliper, after inoculating for 15 days; the mice were divided in two groups. Mice were anesthetized in the induction chamber with 3% isoflurane (filtered (0.2 μm) air at 1 L/min flow rate used as carrier). Anesthetized mice were transferred to the imaging chamber with isoflurane reduced to 2%, and $O_2$ was used as a carrier at 1 L/min flow rate. The mice were imaged by IVIS Spectrum in vivo Imaging System (PerkinElmer, Waltham, MA) after intraperitoneal injection of D-Luciferin (30 mg/mL), 100 μL/20 g body weight. Dynamic scan was performed for 35 min with an image captured every 2 min, the highest luminescence time point was selected as photograph time after injection of D-Luciferin. The first-time picture was set as 0 h of ATP content in vivo. Then, injection of the GCA (10 μM) (Cayman Chemical, Cat#1005036-73-6) in DMSO into the tumor region, or the same amount of DMSO was used as negative control, and imaged again with the same time point after 48 h.

**In vitro experiment for co-cultured cells**. The IL-1β was assayed form BALB/c mouse DCs. The lysate from $5 \times 10^5$ CT26 cells was s.c. injected into BALB/c mice for 2 weeks, and DCs were prepared from the mouse spleen. The DCs were labeled with a CD11c antibody, ligated to Mouse CD11c MicroBeads (Miltenyi Biotec, Cat#130-108-338), and purified by MACS columns (Miltenyi Biotec) according to the manufacturer's instructions. CT26 cells were pre-transfected with sh-Scram or sh-Arf1, or double transfected with sh-Scram or sh-Arf1 with sh-Calr, sh-ERp46, or sh-Hmgb1 plasmids, and the knockdown efficiency was shown by immunoblotting. The sh-Scram (GFP-shRNA) and Arf1-shRNA was purchased from Sigma, as shown in the table. For the experiment, $1 \times 10^5$ CT26 cells were co-cultured with $1 \times 10^4$ to $1 \times 10^5$ DCs for 48 h, and the IL-1β levels in the cell medium were determined by the Mouse IL-1β ELISA Kit (Biolegend, Cat#432601) according to the manufacturer's instructions.

The CD8α+ T cells and CD4 T cells were purified after mice were s.c. injected with the lysate from $5 \times 10^5$ CT26 cells for 2 weeks, and CD8α T cells were purified from the mouse spleen with Mouse CD8α (Ly-2) MicroBeads (Miltenyi Biotec, Cat#130-117-044). The CD4 T cells were purified from mouse spleen with Mouse CD4 (L3T4) MicroBeads (Miltenyi Biotec, Cat#130-117-043). The cell purity was confirmed by FACS. For the experiment, $1 \times 10^6$ CT26 cells transfected with different sh-RNAs were co-cultured with $1 \times 10^5$ DCs and $1 \times 10^5$ CD8α+ T cells or CD4 T cells for 48 h, and the IFNγ level in the cell culture medium was determined using the Mouse IFNγ ELISA Kit (Biolegend, Cat#430801) according to the manufacturer's instructions.

**In vitro ATP and HMGB1 assay**. The extracellular ATP was measured by the ENLITEN® ATP Assay System (Promega, Cat#FF2000). The extracellular HMGB1 was measured by the Human or Mouse HMGB1 ELISA Kit (Abclonal, Cat#EHH0016) according to the manufacturer's instructions. In this experiment, $1 \times 10^6$ Huh-7 cells or $5 \times 10^5$ CT26 cells were cultured in RPMI164 medium (10% FBS, 100 units/mL penicillin/streptomycin, 2.0 mM L-glutamine, 50 μg/mL normocin, and 2.0 g/L sodium bicarbonate) overnight, and the cells were treated the next day with 10 μM BFA or 5 μM GCA. After 48 h of treatment, the cell medium was collected, cleared of dye by centrifugation at $800 \times g$ for 5 min, and assayed for ATP and HMGB1 content according to the manufacturer's instructions.

**Enzyme-linked immunosorbent assays**. The mouse alanine transaminase (ALT), IL-1β, and IFNγ assays were performed with the Alanine Transaminase Colorimetric Activity Assay Kit (Cayman, Cat#700260), Mouse IL-1β ELISA Kit (Biolegend, Cat#432601), and the Mouse IFNγ ELISA Kit (Biolegend, Cat#430801) according to the manufacturer's instructions.

**Mitochondrial and glycolysis stress assays**. Mitochondrial stress was assayed by the Seahorse XFp Cell Mito Stress Test Kit (Agilent, Cat#103010-100), and the glycolysis stress assay was performed by the Seahorse XFp Glycolysis Stress Test Kit (Agilent, Cat#103017-100) according to the manufacturer's instructions.

**Human cancer analysis**. For human samples, the association between Arf1 expression and patient survival was analyzed by the Kaplan–Meier scanner using the R2 platform or TCGA dataset database analyzed by UCSC Xena. The cutoff was set to the default setting. For the analysis of the Arf1 expression level versus (vs) the T-cell infiltration of different kind of tumors, the CD8α expression level was used to assess the infiltration of cytotoxic T cells, and Spearman's correlations were used to calculate the Arf1 expression vs T-cell infiltration. The different expression of Arf1 levels between a human patient's tumor and the adjacent normal tissues across the TCGA database was analyzed by the TIMER (https://cistrome.shinyapps.io/timer/) online system[47]. The statistical significance of differential expressions was evaluated using the Wilcoxon test. The ARF1 expression levels vs IL-1β and IFNγ were analyzed in The Cancer Genome Atlas dataset (NCI, NIH) by the Cbioportal website (http://www.cbioportal.org/) using Spearman's correlations.

**Quantification and statistical analysis**. Please refer to the figure legends for experimental details including the descriptions of samples (cells, mice) and statistical details. The number of mice used is shown in the figures, and the cell

experiments were performed with triplicate samples and in two independent experiments. Data were plotted and analyzed by GraphPad prism 8.0 software (GraphPad Software). Data are shown as the mean ± SEM. Differences were analyzed using two-tailed Student's $t$-tests and one-way ANOVA (Bonferroni posttest) and were considered significant when the $p$-value ≤ 0.05.

**Reporting Summary**. Further information on research design is available in the Nature Research Reporting Summary linked to this article.

## Data availability

All the data supporting the findings of this study are available within the article and its supplementary information files and from the corresponding author upon reasonable request. A reporting summary for this article is available as a Supplementary Information file. The source data underlying Figures and Supplementary Figures are provided as a Source Data file.

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

## Acknowledgements

The authors thank Dr. Lino Tessarollo lab and Eileen Southon for assistance with the blastocyst injections; Helen Shin for mouse genotyping; the Pathology/Histotechnology Laboratory, NCI-Frederick, for tissue sectioning and staining; Amy Minz for help with the mouse experiments; Kim Klarmann and the Frederick Flow Cytometry Core for help with the flow cytometry; and Kunio Nagashima and The Electron Microscopy Laboratory (EML) for help with EM experiments. This research was supported by the Intramural Research Program of the National Institutes of Health (NIH), NCI, Center for Cancer Research.

## Author contributions

S.X.H. and G.W. conceived and designed the experiments. G.W. and J.X. performed the experiments. J.Z., W.Y., D.L. and W.J.C. assisted with experiments. S.X.H., G.W. and W.J.C. analyzed the data. S.X.H. and G.W. wrote the manuscript.

## Competing interests

The authors declare no competing interests.
