## [Transparent Peer Review File · Nature Communications]

Reviewers' comments:

Reviewer #1 (Remarks to the Author):

In their manuscript, Wang et al. set out to investigate the role of Arf1 depletion in cancer stem cells of murine cancers and human cell line xenografts. With the studies using *Drosophila melanogaster* as a model organism, the group has previously shown that Arf1-mediated lipolysis pathway specifically sustains stem cells as the knockdown of this gene disrupts lipid metabolism, induces lipid droplet accumulation and stem cell necrosis in the fruit flies. These results encouraged the authors to investigate the importance of Arf1 also in mammalian systems. They first demonstrate that conditional knock-out of Arf1 in inducible murine colonic/intestinal tumor model (Axin2-CreER/APCfl/fl/Arf1fl/fl mouse model) ablate the Axin2-expressing cells and reduce the tumor burden in vivo. More detailed structural analysis of Arf1-mutant murine intestines showed aberrations in mitochondrial architecture and cellular respiration. Additionally, they noticed that Arf1-depleted intestinal tumors displayed accumulation of certain type of immune cells (dendritic cells and selection of T-lymphocytes) and increased expression of some immune cell activation markers. Upon these results, the authors concluded that the loss of Arf1 in stem cells triggers T cell infiltration and activation, leading to cancer stem cell death and prolonged survival of the mice.

The authors then turned their interest and subsequently focused their studies on more lipid-rich organ – liver. They analyzed the role of Arf1 depletion in both developing and adult murine livers and found that while adult hepatocytes are not affected by the gene knock-out, Arf1 deletion during stem-cell rich early liver development resulted in underdeveloped livers with embryonic lethality. They further studied the anti-tumor effect of Arf1 inhibition in a mouse hepatocellular liver tumor model induced by overexpression of MYC proto-oncogene. They found that administration of Arf1 inhibitors (GCA and BFA) reduced the number of liver tumors and extended the lifespan of the mice. Arf1 depletion in Axin2-expressing cells of liver additionally showed decrease in number of liver tumors and, similarly to intestinal cancer model, the increased number of certain immune cell types was also observed in Arf1-mutant liver cancers. Next, the authors hypothesized that the anti-tumor effect of Arf1 ablation in mouse intestinal and liver tumors might depend on the presence of CD4+ and CD8+ T cells. Therefore, they depleted T-lymphocytes after tumor formation by injecting anti-CD4 or anti-CD8 antibodies to deplete the respective T cells and observed increase in tumor burden and reduction in lifespan of these mice. To support the hypothesized mechanism of action of Arf1, Arf-inhibition was shown to trigger ER-stress, induce damage-associated molecular patterns (DAMPs) and dendritic cell infiltration in human cell line xenografts. Arf1 inhibition was shown to induce anti-tumor immune responses through DAMPs. Lastly, the authors experimented with a mouse vaccination model which allowed them to demonstrate that Arf1-ablated cells protect animals from developing tumors. TCGA dataset analysis confirmed that Arf1 is frequently amplified or overexpressed in different cancer types and Arf1-low group had a significantly better survival rate than the Arf1-high group.

MAJOR REMARKS:

1) Authors make hasty generalizations about the role of Arf1 in stem cells that is not supported by the presented data. Although I agree that there is quite clear effect of Arf1 depletion on tumor volume and aggressiveness, the overall conclusion that Arf1 specifically sustains cancer stem cells is not entirely supported by the currently presented evidence. Correct rephrasing is required.

ADDITIONAL REMARKS:

1) Authors claim that Arf1-depletion specifically ablates stem cells. They use two main Cre-mediated stem cell-targeting models: Axin2-CreER/R26RLacZ (liver and intestine) and Lgr5-EGFP-IRES-CreERT2 (intestine). They consider Axin2 to be a stem cell marker in both liver and intestinal epithelial cells and, indeed, observe loss of Axin2-LacZ signal upon Arf1 knock-out in Axin+ cells in both epithelia (Figure 1a, Supplementary Figure 4b). In intestine, Lgr5 is considered the most reliable marker of adult stem cells, however, in intestinal tumor model (Lgr5/Arf1fl/fl/Apcfl/fl) the expression of Lgr5 (evident from GFP signal, Figure 1b) seems to be reduced but not completely

lost in Arf1 mutant mice. This observation would argue with the main claim that Arf1 depletion specifically ablates the stem cells. How to explain?

2) Line 195: "Axin2-CreER is selectively expressed in liver stem cells". In normal homeostatic liver conditions the tissue has a very slow turnover. Axin2+ hepatocytes have been reported to have self-renewal capacity in homeostatic livers (Wang et al. 2015), however, these cells have very low proliferation rate and are only able to replace on average 40% of hepatocytes within liver and doesn't account for cholangiocyte renewal which suggests the presence of other stem cell populations in liver. Notably, liver has a remarkable regeneration capacity and is able to restore the original size upon injury. During injury conditions different bipotent stem cell population appears within liver, characterized by Lgr5 stem cell marker expression and periportal location. These Lgr5+ cells can give rise to both biliary and hepatocyte lineages. Thus, the above claim that Axin2 selectively marks liver stem cells can be considered misleading and essentially wrong.

3) It would be important to know if the liver regeneration capacity (stem cell pool) is influenced by Arf1 knock-out upon liver injury. Therefore, it would be a great addition if authors could perform some liver injury experiments.

4) In homeostatic livers, where the turnover rate is very slow, authors specifically deplete Arf1 in Albumin-expressing hepatocytes using Albumin-Cre reporter mice. With these experiments they conclude that in short-term knockdown of Arf1 in differentiated liver cells has no effect, however, they don't show the long-term effect which would be more important considering liver has a slow turnover. In contrast to liver, intestinal epithelium has a high turnover rate and the renewal is constantly fuelled by Lgr5-positive stem cells at the crypt bottoms. The authors do not characterize very well the effect of Arf1-depletion on normal intestinal turnover. Authors claim to ablate stem cells, yet the integrity of the crypts and overall survival of the mice do not seem to be affected in Lgr5/Arf1fl/fl mice. How to explain?

5) To confirm the mechanistic link between the Arf1-depletion and immune cell infiltration, authors use anti-CD4 and anti-CD8 antibodies to support their hypothesis. The better control system would be to use immunodeficient mice for that experiment. Observing better tumor growth in immunodeficient mice would be the best proof for such hypothesis.

6) In the original Barker et al. (2008) paper the survival of Lgr5-EGFP-IRES-creERT2/APCflox/flox mice have been reported to be 36 days, however, in this manuscript the comparable Lgr5/APCfl/fl mice survive only up to 20 days and Lgr5/APCfl/fl/Arf1fl/fl mutants up to 32 days. How to explain?

7) At several location in the manuscript authors use Arf11 instead of Arf1. Please correct.

8) In regards to the tumor analysis, authors often mention "tumor size/number" to describe the tumors, however no measurements or quantification of tumor sizes or numbers, respectively, have been given to support that claim.

9) The differential usage of various human cancer cell lines for different experiments is not very well reasoned/explained.

10) Some newly used terms have not been explained (AML blasts, HSCs...)

Reviewer #2 (Remarks to the Author):

In this manuscript, Wang G et al. elaborated that Arf1-mediated lipid metabolism sustains CSC and its ablation induces antitumor immune response in mice through induction of mitochondrial defects, ES stress and necrosis of CSC. This is a well-designed and well-written manuscript. However, there are several questions need to be addressed.

1. From the manuscript, it is suggested that Arf1 has significant function in CSCs rather than in mature cells. However, there is extensive data in this manuscript that is generated from different cell lines without information of percentage of CSCs in these cell lines, rather than from CSCs directly. Since the authors are trying to marry Arf1 with CSCs in terms of improving therapeutic efficacy with treatment, it would be critical to provide the information as abovementioned.

2. The authors have emphasizes that the ablation Arf1 induces stem cell necrosis that was evidenced in the Drosophila study. However, there is no evidence or data in this manuscript to indicate the

same process occurs in mice rather than apoptosis.

3. It is interesting to see the expression of PD-L1 is down-regulated by Arf1 inhibition or ablation. It would be relevant to elaborate more regarding the mechanism if Arf1 regulates PD-L1 expression.

4. NK T cells are critical to control liver cancer tumorigenesis, probably equal important compared to CD4 or CD8 cells. Do the authors have this relevant data?

5. In Supplemental Figure S21, the authors found out the Arf1 deletion with Foxa3-Cre resulted in significant reduction of markers of hepatoblast, hepatocyte, cholangiocytes and progenitor as well. What is the time point of the sample collection of embryo stage? Is the term of hepatoblast interchangeable with progenitor in this case?

Reviewer #3 (Remarks to the Author):

Wang et al 2019

Review

Date 17/07/2019

Title: Arf1-mediated lipid metabolism sustains cancer stem cells and its ablation induces anti-tumour immune responses in mice

The manuscript by Wang and colleagues comprehensively investigates the role Arf1 plays in sustaining cancer stem cell populations and disease aggressiveness. Using various murine models they show that Arf1 ablation influences immune infiltration and tumour clearance. The authors demonstrate Arf1 loss results in DAMP release, potentially as a result of ER-stress and an inhibition of protein translation. I commend the authors on the body of work but they have not explored fully the impact on lipid metabolism, which I will discuss below.

The primary phenotype observed appears to be the impact of Arf1 loss on protein transition between the golgi and the ER. The increase in lipid droplet number in response to BFA and GCA could be a result of increased lipid de novo synthesis, increased lipid uptake or reduced fatty acid oxidation (FAO). The authors state that basal OCR and maximal capacity are reduced but perhaps mitochondria are switching to a FAO phenotype rather than a glycolytic one. Although the authors cite the literature, this is not explored in the manuscript but is shown in the diagram in the supplemental data. The authors should investigate whether there are changes in FASN, CPT1, CD36 expression in their models of Arf1 loss to give more clarification on the mechanism of lipid droplet accumulation. If the models are switching to a FAO phenotype then they will be more sensitive to etomoxir at CPT1A inhibitor, this should be tested.

The authors also do not explore the content of the lipid droplets. Once these cells are ruptured as a result of Arf1 inhibition they could be releasing lipids, including inflammatory lipids, into the tumour microenvironment complementing DAMPs in promoting an inflammatory response. The authors should perform lipidomics on the MYC-On tumours and Lgr5/Arf1f/f/Apcf/f tumours to get a quantitative understanding of the lipid environment. This would be particularly pertinent in the MYC-On setting where Myc is a known regulator of de novo lipogenesis (Eberlin et al 2014 Alteration of the lipid profile in lymphomas induced by MYC overexpression). It will also inform the authors of changes in organelle membrane fluidity, such as cardiolipin changes, which impact complex OXPHOS activity, and the saturated/mono/polyunsaturated fatty acid levels in these tumours (you can differentiate between free fatty acid and PE, PS, PI, CL etc).

In this vein, if the authors believe that the phenotype revolves around ER-stress promoting immune infiltration, in the MYC-On model treated with BFA or GCA, would treatment with

thapsigardin or tunicamycin further reduce tumour number and increase animal survival? This would strengthen their findings that it is perturbations in protein synthesis and transit that is causing this phenotype.

The authors reply on transcriptional impact on PD-L1 expression. They should histologically investigate protein expression and localisation as this would be more informative. Classifying tumours on their PD-L1 status i.e. scored by a pathologist would give weight to this observation being able to be also observed in patients.

I want to reiterate that I think this a comprehensive and well delivered study, but in my opinion the link with lipid metabolism at the moment is not as convincing as the rest of the data, which is very impressive.

Minor corrections:

Line 49: define "HSCs"

Line 69: "kill two birds with one stone" is not suitable language, change for "multimodal" or an equivalent

Line 82: "achieve a more potent effect"

Line 109: define BFA and GCA

Line 113: replace "broken and swollen" with aberrant or irregular mitochondria or an equivalent

Line 142: delete "in"

Line 329: define BPA

Line: 392: see above Line 69

Reviewers' comments:

Reviewer #1 (Remarks to the Author):

In their manuscript, Wang et al. set out to investigate the role of Arf1 depletion in cancer stem cells of murine cancers and human cell line xenografts. With the studies using *Drosophila melanogaster* as a model organism, the group has previously shown that Arf1-mediated lipolysis pathway specifically sustains stem cells as the knockdown of this gene disrupts lipid metabolism, induces lipid droplet accumulation and stem cell necrosis in the fruit flies. These results encouraged the authors to investigate the importance of Arf1 also in mammalian systems. They first demonstrate that conditional knock-out of Arf1 in inducible murine colonic/intestinal tumor model (Axin2-CreER/APC^{fl/fl}/Arf1^{fl/fl} mouse model) ablate the Axin2-expressing cells and reduce the tumor burden in vivo. More detailed structural analysis of Arf1-mutant murine intestines showed aberrations in mitochondrial architecture and cellular respiration. Additionally, they noticed that Arf1-depleted intestinal tumors displayed accumulation of certain type of immune cells (dendritic cells and selection of T-lymphocytes) and increased expression of some immune cell activation markers. Upon these results, the authors concluded that the loss of Arf1 in stem cells triggers T cell infiltration and activation, leading to cancer stem cell death and prolonged survival of the mice.

The authors then turned their interest and subsequently focused their studies on more lipid-rich organ – liver. They analyzed the role of Arf1 depletion in both developing and adult murine livers and found that while adult hepatocytes are not affected by the gene knock-out, Arf1 deletion during stem-cell rich early liver development resulted in underdeveloped livers with embryonic lethality. They further studied the anti-tumor effect of Arf1 inhibition in a mouse hepatocellular liver tumor model induced by overexpression of MYC proto-oncogene. They found that administration of Arf1 inhibitors (GCA and BFA) reduced the number of liver tumors

and extended the lifespan of the mice. Arf1 depletion in Axin2-expressing cells of liver additionally showed decrease in number of liver tumors and, similarly to intestinal cancer model, the increased number of certain immune cell types was also observed in Arf1-mutant liver cancers. Next, the authors hypothesized that the anti-tumor effect of Arf1 ablation in mouse intestinal and liver tumors might depend on the presence of CD4⁺ and CD8⁺ T cells. Therefore, they depleted T-lymphocytes after tumor formation by injecting anti-CD4 or anti-CD8 antibodies to deplete the respective T cells and observed increase in tumor burden and reduction in lifespan of these mice. To support the hypothesized mechanism of action of Arf1, Arf1-inhibition was shown to trigger ER-stress, induce damage-associated molecular patterns (DAMPs) and dendritic cell infiltration in human cell line xenografts. Arf1 inhibition was shown to induce anti-tumor immune responses through DAMPs. Lastly, the authors experimented with a mouse vaccination model which allowed them to demonstrate that Arf1-ablated cells protect animals from developing tumors. TCGA dataset analysis confirmed that Arf1 is frequently amplified or overexpressed in different cancer types and Arf1-low group had a significantly better survival rate than the Arf1-high group.

MAJOR REMARKS:

1) Authors make hasty generalizations about the role of Arf1 in stem cells that is not supported by the presented data. Although I agree that there is quite clear effect of Arf1 depletion on tumor volume and aggressiveness, the overall conclusion that Arf1 specifically sustains cancer stem cells is not entirely supported by the currently presented evidence. Correct rephrasing is required.

Reply:

Thank the reviewer's suggestion. We rephrased cancer stem cells as "cells enriched with cancer stem cells".

ADDITIONAL REMARKS:

1) Authors claim that Arf1-depletion specifically ablates stem cells. They use two main Cre-mediated stem cell-targeting models: Axin2-CreER/R26RLacZ (liver and intestine) and Lgr5-EGFP-IRES-CreERT2 (intestine). They consider Axin2 to be a stem cell marker in both liver and intestinal epithelial cells and, indeed, observe loss of Axin2-LacZ signal upon Arf1 knock-out in Axin⁺ cells in both epithelia (Figure 1a, Supplementary Figure 4b). In intestine, Lgr5 is considered the most reliable marker of adult stem cells, however, in intestinal tumor model (Lgr5/Arf1^{fl/fl}/Apc^{fl/fl}) the expression of Lgr5 (evident from GFP signal, Figure 1b) seems to be reduced but not completely lost in Arf1 mutant mice. This observation would argue with the main claim that Arf1 depletion specifically ablates the stem cells. How to explain?

Reply:

Although Lgr5-eGFP-IRES-CreERT2 has been successfully used to delete tumor suppressor *Apc* in intestinal stem cells to generate stem-cell-derived adenomas (Barker N. et al., Nature. 2009), yet its low level expression and its silencing in patches of adjacent crypts have not allowed quantitative gene deletion (Schuijers J. et al., Stem Cell Reports.2014). Therefore, the *Arf1* gene maybe not completely deleted by Lgr5-CreERT2. It is also possible that mouse *Arf1* has a redundant function, another unidentified gene partially compensates *Arf1*-loss-of-function.

2) Line 195: “Axin2-CreER is selectively expressed in liver stem cells”. In normal homeostatic liver conditions the tissue has a very slow turnover. Axin2⁺ hepatocytes have been reported to have self-renewal capacity in homeostatic livers (Wang et al. 2015), however, these cells have very low proliferation rate and are only able to replace on average 40% of hepatocytes within liver and doesn't account for cholangiocyte renewal which suggests the presence of other stem cell populations in liver. Notably, liver has a remarkable regeneration capacity and is able to restore the original size upon injury. During injury conditions different bipotent stem cell population appears within liver, characterized by Lgr5 stem cell marker expression and periportal location. These Lgr5⁺ cells can give rise to both biliary and hepatocyte lineages. Thus, the above claim that Axin2 selectively marks liver stem cells can be considered misleading and essentially wrong.

Reply:

The reviewer is right that the liver stem cells are very complicated. During injury conditions different bipotent stem cell populations have been reported within liver, including de-differentiated hepatocytes, Lgr5⁺ stem cells and etc.. However, only Axin2⁺ liver stem cells have been reported to have self-renewal capacity in homeostatic livers (Wang B. et al. Nature. 2015). Therefore, we deleted *Arf1* using Axin2-CreERT2 in MYC-induced mouse liver tumors and found that deletion of *Arf1* in Axin2⁺ stem cells significantly blocked liver tumor development. Axin2⁺ stem cells may only represent one of many liver stem cells, therefore we revised “Axin2-CreER is selectively expressed in liver stem cells” as “Axin2-CreER is selectively expressed in one type of liver stem cells or Axin2⁺ liver stem cells”.

3) It would be important to know if the liver regeneration capacity (stem cell pool) is influenced by *Arf1* knock-out upon liver injury. Therefore, it would be a great addition if authors could perform some liver injury experiments.

Reply:

We have performed the reviewer suggested experiment. We used the Acetaminophen (APAP) (300mg/kg) to induce the liver injury in control and Alb-Cre/*Arf1*^{fl/fl} mice. We found that the liver regeneration capacity was reduced once *Arf1* was knocked out in hepatocytes (Alb-Cre/*Arf1*^{fl/fl}) in comparison with that of control mice. It was reported that de-differentiation of hepatocytes

was involved in injury-induced liver regeneration and Arf1 may play a role in that process. We will follow this initial observation in the future experiments.

The experimental results are shown below and also incorporated into the revised manuscript:

Figure S3A-C. Arf1 knockdown in mouse liver results in delay hepatocytes regeneration. (A) H&E stained different time points of control (Alb-Cre/Arf1^{f/+}) or Arf1 knockdown (Alb-Cre/Arf1^{f/f}) mouse livers after treatment with APAP. (B) Nitrotyrosine stained of control (Alb-Cre/Arf1^{f/+}) or Arf1 knockdown (Alb-Cre/Arf1^{f/f}) mouse livers after treatment with APAP (300 mg/kg). (C) ELISA of ALT level, quantification of pathology score and quantification of Nitrotyrosine positive cells in the control (Alb-Cre/Arf1^{f/+}) or Arf1 knockdown (Alb-Cre/Arf1^{f/f}) mouse livers after treatment with APAP. Scale bar as 200 μ m (A) and 50 μ m (B). Data are shown as the mean \pm SEM. * p <0.05, ** p <0.01 and *** p <0.001 by Student's t test.

4) In homeostatic livers, where the turnover rate is very slow, authors specifically deplete Arf1 in Albumin-expressing hepatocytes using Albumin-Cre reporter mice. With these experiments they conclude that in short-term knockdown of Arf1 in differentiated liver cells has no effect, however, they don't show the long-term effect which would be more important considering liver has a slow turnover. In contrast to liver, intestinal epithelium has a high turnover rate and the renewal is constantly fueled by Lgr5-positive stem cells at the crypt bottoms. The authors do not characterize very well the effect of Arf1-depletion on normal intestinal turnover. Authors claim to ablate stem cells, yet the integrity of the crypts and overall survival of the mice do not seem to be affected in Lgr5/Arf1^{f/f} mice. How to explain?

Reply:

We have kept the Alb-Cre/Arf1^{f/f} mice for more than one and a half year and haven't found any difference between control and Arf1-knockout mice, regarding to the body weight and physical conditions. As described above, although Lgr5-eGFP-IRES-CreERT2 has been successfully used to delete tumor suppressor Apc in intestinal stem cells to generate stem-cell-derived adenomas (Barker N. et al., 2009), yet its low level expression and its silencing in patches of adjacent crypts have not allowed quantitative gene deletion (Schuijers J. et al., 2014). In this study, we only used Lgr5-CreERT2 to study Arf1's function in Apc-deletion-induced stem cell tumors. We will study Arf1's function in normal intestinal stem cells using other more effective Cre (such as Olfm4-IRES-eGFP-CreERT2) in future experiments.

5) To confirm the mechanistic link between the Arf1-depletion and immune cell infiltration, authors use anti-CD4 and anti-CD8 antibodies to support their hypothesis. The better control system would be to use immunodeficient mice for that experiment. Observing better tumor growth in immunodeficient mice would be the best proof for such hypothesis.

Reply:

This is a logic and reasonable suggestion. Actually, we already crossed Lgr5-CreER/Arf1^{f/f}/Apc^{f/f} mice with Rag1-KO and IFNg-KO mice before submitting the manuscript. We have recently gotten the homozygous mice of Lgr5-CreER/Arf1^{f/f}/Apc^{f/f}/Rag1-KO and Lgr5-CreER/Arf1^{f/f}/Apc^{f/f}/IFNg-KO mice and performed the suggested experiments. After giving 3 tamoxifen (1 mg TAM/10 g mice) injections, we monitored tumor growth and survival times of the experimental mice. In comparison with Lgr5/Apc^{f/f} and Lgr5/Arf1^{f/f}/Apc^{f/f} mice, we found that with Rag1 and IFNg knockouts (particularly Rag1 knockout), the mice survival times were significantly decreased than those of Lgr5-CreER/Arf1^{f/f}/Apc^{f/f} mice, but were still somehow longer than those of Lgr5/Apc mice. The tumor numbers in the Lgr5-CreER/Arf1^{f/f}/Apc^{f/f}/IFNg-KO and Lgr5-CreER/Arf1^{f/f}/Apc^{f/f}/Rag1-KO also significantly increased in comparison with those in the Lgr5-CreER/Arf1^{f/f}/Apc^{f/f} mice. Therefore, these results further support our conclusion that the anti-tumor effects of Arf1 ablation are through inducing anti-tumor immune responses.

Figure 4G-I. (A) Immunohistochemical staining of GFP in the indicated genotypes. (B-C) Percent of survival (B) and intestine tumor number (C) in the indicated mice. Data are shown as the mean \pm SEM. * $p < 0.05$, ** $p < 0.01$, *** $p < 0.001$ by Student's t test.

6) In the original Barker et al. (2009) paper the survival of Lgr5-EGFP-IRES-creERT2/APC^{flox/flox} mice have been reported to be 36 days, however, in this manuscript the comparable Lgr5/APC^{f1/f1} mice survive only up to 20 days and Lgr5/APC^{f1/f1}/Arf1^{f1/f1} mutants up to 32 days. How to explain?

Reply:

In Barker et al. (2009) paper, they did one-time tamoxifen injection. In our experiments, we used 3 time tamoxifen (1 mg TAM/10 g mouse) injections, which may explain the reason that the mice grew tumors more quickly in our experiments. In the original figure legend, we made a mistake and wrote one tamoxifen injection. In the revised manuscript, we changed to 3 time Tamoxifen injections.

7) At several location in the manuscript authors use Arf1 1 instead of Arf1. Please correct.

Reply:

Thank the reviewer, we corrected the error.

8) In regard to the tumor analysis, authors often mention “tumor size/number” to describe the tumors, however no measurements or quantification of tumor sizes or numbers, respectively, have been given to support that claim.

Reply:

We re-quantified the tumor numbers or tumor sizes in the revised manuscript.

9) The differential usage of various human cancer cell lines for different experiments is not very well reasoned/explained.

Reply:

The 4 cell lines used in our study are mouse tumor cell lines, CT26 colon carcinoma, 4T1 breast carcinoma, A20 B cell lymphoma and B16-10 melanoma models. The main reason that we selected these cell lines is that they can induce tumor formation in wild-type mice and have been successfully used in anti-tumor immune studies (see, e.g., Langowski JL. et al., Nature.2006; Sagiv-Barfi I. et al., Sci. Transl. Med. 2018).

10) Some newly used terms have not been explained (AML blasts, HSCs...)

Reply:

AML blasts and HSCs have been explained in the revised manuscript. AML--the acute myeloid leukemia blast (AML blasts), HSCs-- hematopoietic stem cells (HSCs).

Reviewer #2 (Remarks to the Author):

In this manuscript, Wang G et al. elaborated that Arf1-mediated lipid metabolism sustains CSC and its ablation induces antitumor immune response in mice through induction of mitochondrial defects, ES stress and necrosis of CSC. This is a well-designed and well-written manuscript. However, there are several questions need to be addressed.

1. From the manuscript, it is suggested that Arf1 has significant function in CSCs rather than in mature cells. However, there is extensive data in this manuscript that is generated from different cell lines without information of percentage of CSCs in these cell lines, rather than from CSCs directly. Since the authors are trying to marry Arf1 with CSCs in terms of improving therapeutic efficacy with treatment, it would be critical to provide the information as abovementioned.

Reply:

This is an important suggestion. We have analyzed CSC numbers in CT26, B16-F10 and 4T1 cells, using Flow cytometry with two reported CSC markers, CD133 and CD44 (Klonisch T. et al., Trends in Molecular Medicine. 2008). We found that CT26 cells have about 98.3% CD133⁺CD44^{Hi} cancer stem cells, B16-F10 cells have about 40.1% CD133⁺CD44^{Hi} cancer stem cells, and 4T1 cells have about 76.7% CD133⁺CD44^{Hi} cancer stem cells. The data were shown below and also incorporated into the revised manuscript Figure S9E.

Figure S9E. Flow cytometry analysis the CD133+ and CD44+ stem cells in the CT26, B16F10 and 4T1 cells.

2. The authors have emphasizes that the ablation Arf1 induces stem cell necrosis that was evidenced in the Drosophila study. However, there is no evidence or data in this manuscript to indicate the same process occurs in mice rather than apoptosis.

Reply:

First, there is no significant difference of cleaved Caspase3 in the Arf1-KO (*Foxa3-Cre/Arf1^{ff}*) mouse embryonic livers in comparison with the control mouse livers (Figure below), suggesting that hepatoblasts didn't die from apoptosis.

Second, our EM data (supplemental Figure 2L) suggest that hepatoblasts in *Foxa3-Cre/Arf1^{ff}* mice died from necrosis (ruptured membrane with intact nuclei) (our EM expert told us that these were necrosis).

Figure S7A. Western blot of Caspase3 and cleaved caspase 3 in the *Foxa3-Cre/Arf1^{+/+}* and *Foxa3-Cre/Arf1^{ff}* mice.

3. It is interesting to see the expression of PD-L1 is down-regulated by Arf1 inhibition or ablation. It would be relevant to elaborate more regarding the mechanism if Arf1 regulates PD-L1 expression.

Reply:

Because the PD-L1 can be regulated by HIF1-a, Myc, STATs, NF-kB and AP-1(Wang Y. et al., Front Pharmacol.2018; Zerdes I. et al., Oncogene. 2018), we checked expression of these transcription factors using qRT-PCR in the control and GCA-treated MYC-ON mice and found that only AP-1/C-Jun was decreased in GCA-treated MYC-ON mice in comparison with that in control mice (Figure below) and also incorporated into the revised manuscript Figure S6E.

Therefore, Arf1 inhibition may down-regulate PD-L1 through reducing AP-1/C-Jun signaling pathway.

Figure S6E. qRT-PCR amplify the transcription factors in the liver of DMSO or GCA treated MYC-ON mice. Data are shown as the mean \pm SEM. * $p < 0.05$, ** $p < 0.01$, *** $p < 0.01$ by Student's t test. Scale bars are as indicated.

4. NK T cells are critical to control liver cancer tumorigenesis, probably equal important compared to CD4 or CD8 cells. Do the authors have this relevant data?

Reply:

Thanks for the important suggestion, we checked the NK T cells in the liver cancer of Axin2-CreER/Arf1^{f/f}/Cebpb/Myc mice by FACS and didn't find significant change (Figure below) and also incorporated into the revised manuscript Figure S5D.

Figure S4D. Flow cytometric analysis of immune cells in the liver of mice with the indicated genotypes. Data are shown as the mean \pm SEM. *p<0.05, **p<0.01 by Student's t test. Scale bars are as indicated.

5. In Supplemental Figure S2I, the authors found out the *Arf1* deletion with *Foxa3-Cre* resulted in significant reduction of markers of hepatoblast, hepatocyte, cholangiocytes and progenitor as well. What is the time point of the sample collection of embryo stage? Is the term of hepatoblast interchangeable with progenitor in this case?

Reply:

The *Foxa3-Cre/Arf1* mice die between E13.5 and E15.5 of the embryo stage. We collected the tissues at ~E13.5. At this stage the hepatoblasts start differentiating into hepatocytes and cholangiocytes (Miriam Gordillo, Development. 2015). In the mouse liver bud, hepatoblasts are the only epithelial progenitor cells of fetal liver, therefore hepatoblast and progenitor are interchangeable.

Reviewer #3 (Remarks to the Author):

Wang et al 2019

Review

Date 17/07/2019

Title: Arf1-mediated lipid metabolism sustains cancer stem cells and its ablation induces anti-tumour immune responses in mice

The manuscript by Wang and colleagues comprehensively investigates the role Arf1 plays in sustaining cancer stem cell populations and disease aggressiveness. Using various murine models they show that Arf1 ablation influences immune infiltration and tumor clearance. The authors demonstrate Arf1 loss results in DAMP release, potentially as a result of ER-stress and an inhibition of protein translation. I commend the authors on the body of work but they have not explored fully the impact on lipid metabolism, which I will discuss below.

1. The primary phenotype observed appears to be the impact of Arf1 loss on protein transition between the golgi and the ER. The increase in lipid droplet number in response to BFA and GCA could be a result of increased lipid de novo synthesis, increased lipid uptake or reduced fatty acid oxidation (FAO). The authors state that basal OCR and maximal capacity are reduced but perhaps mitochondria are switching to a FAO phenotype rather than a glycolytic one. Although the authors cite the literature, this is not explored in the manuscript but is shown in the diagram in the supplemental data. The authors should investigate whether there are changes in FASN, CPT1, CD36 expression in their models of Arf1 loss to give more clarification on the mechanism of lipid droplet accumulation. If the models are switching to a FAO phenotype then they will be more sensitive to etomoxir at CPT1A inhibitor, this should be tested.

Reply:

We examined expression of FASN, CPT1A, CD36 by western blot and qRT-PCR using liver with Arf1 knockout (in the *Foxa3-cre/Arf1^{fl/fl}* mice) and found that Arf1 knockout decreased CPT1A expression but didn't affect expressions of FASN and CD36 (Figure below and also incorporated into the revised manuscript), suggesting that mitochondria didn't switch to a FAO phenotype after Arf1 loss.

Figure S2. (A-B) Western blot (A) and qRT-PCR (B) of indicated protein and genes. Data are shown as the mean \pm SEM. *** $p < 0.001$ by Student's t test. Scale bars are as indicated.

2. The authors also do not explore the content of the lipid droplets. Once these cells are ruptured as a result of Arf1 inhibition they could be releasing lipids, including inflammatory lipids, into the tumour microenvironment complementing DAMPs in promoting an inflammatory response. The authors should perform lipidomics on the MYC-On tumors and *Lgr5/Arf1^{f/f}/Apc^{f/f}* tumors to get a quantitative understanding of the lipid environment. This would be particularly pertinent in the MYC-On setting where Myc is a known regulator of de novo lipogenesis (Eberlin et al 2014 Alteration of the lipid profile in lymphomas induced by MYC overexpression). It will also inform the authors of changes in organelle membrane fluidity, such as cardiolipin changes, which impact complex OXPHOS activity, and the saturated/mono/polyunsaturated fatty acid levels in these tumours (you can differentiate between free fatty acid and PE, PS, PI, CL etc).

Reply:

This is a very good suggestion. We agree with the reviewer that DAMPs maybe not the only mediator that promotes an inflammatory response in Arf1-ablated animal, inflammatory lipids and even mitochondrial DNA-cGAS-STING pathway may also involve in the response. Exploration of all these possibilities is beyond the scope of this paper. We will continue to investigate the other possibilities in our future experiments.

As the first step, we analyzed total RNA using RNA-seq from *Lgr5/Apc^{f/f}* and *Lgr5/Arf1^{f/f}/Apc^{f/f}* mouse intestine tumors and found that 618 genes were up- and down-regulated greater than 2-fold in *Lgr5/Arf1^{f/f}/Apc^{f/f}* mice in comparison with those in *Lgr5/Apc^{f/f}* mice. Among them 49 genes are related to lipid metabolism (Figure below). We will investigate the biological significance of these changes in our future experiments.

Figure. (A) PCA plot of RNA-Seq results from *Lgr5/Arf1^{f/f}/Apc^{f/f}* and *Lgr5/Apc^{f/f}* mice. (B) Up- and down-regulation genes (> 2-fold) in the intestine cancer stem cells of *Lgr5/Arf1^{f/f}/Apc^{f/f}* vs *Lgr5/Apc^{f/f}* mice. (C) Relative to lipid metabolism difference expressed genes (>2 fold) in the *Lgr5/Arf1^{f/f}/Apc^{f/f}* mice compare with *Lgr5/Apc^{f/f}* mice (N=4 mice of each group). (D) Enriched ontology cluster of lipid metabolism genes in (C) by metacape website.

3. In this vein, if the authors believe that the phenotype revolves around ER-stress promoting immune infiltration, in the MYC-On model treated with BFA or GCA, would treatment with thapsigardin or tunicamycin further reduce tumor number and increase animal survival? This would strengthen their findings that it is perturbations in protein synthesis and transit that is causing this phenotype.

Reply:

We treated MYC-ON mice with tunicamycin (2mg/kg) plus BFA (0.5mg/kg) and found that co-treatment with the two chemicals further decreased tumor weights and tumor numbers in the

MYC-ON mice in comparison with those in BFA-treated mice. Treatment alone with tunicamycin did not affect tumor growth in the MYC-ON mice.

Figure S11M-N. (A) surface tumor of MYC-ON mice treated with BFA, Tunicamycin, or BFA+Tunicamycin. (B) liver weight and liver tumor number in the (A). Data are shown as the mean \pm SEM. * p <0.05, ** p <0.01, *** p <0.001 by Student's t test.

4. The authors reply on transcriptional impact on PD-L1 expression. They should histologically investigate protein expression and localization as this would be more informative. Classifying tumours on their PD-L1 status i.e. scored by a pathologist would give weight to this observation being able to be also observed in patients.

Reply:

We stained the tumor tissues with anti-PD-L1 antibody and found that the PD-L1 expressions in the tumor cell surface were reduced after Arf1 knockdown in comparison with those in control (Figure below and also incorporated into the revised manuscript).

We divided levels of PD-L1 expression into three different groups on the MYC-ON mouse tumors as well as tumors with CT26 and 4T1 cell transplantations. I: no stained or weakly stained; II: moderately stained; III: strongly stained. We divided levels of PD-L1 expressions on tumors of the Lgr5/Arf1/Apc mice into two groups. I: weakly or moderately stained, II: strongly stained. The PD-L1 levels were significantly reduced after Arf1 knockdown or Arf1 inhibition with its inhibitors in comparison with those in the corresponding controls.

Figure S6. Immunofluorescence (IF) stained of PD-L1 in the difference tumor models. (A-D) IF stained and quantification of PD-L1 level in the intestine of Lgr5/Apc^{f/f} and Lgr5/Arf1^{f/f}/Apc^{f/f} mice (A), liver of MYC-ON mice (B), transplant CT26 tumor (C) and transplant 4T1 tumor (D).

5. I want to reiterate that I think this a comprehensive and well delivered study, but in my opinion the link with lipid metabolism at the moment is not as convincing as the rest of the data, which is very impressive.

Reply:

We appreciate the reviewer's suggestion. We will further investigate lipid metabolism in Arf1-ablated animal in our future experiments.

Minor corrections:

Line 49: define "HSCs"

Line 69: "kill two birds with one stone" is not suitable language, change for "multimodal" or an equivalent

Line 82: "achieve a more potent effect"

Line 109: define BFA and GCA

Line 113: replace "broken and swollen" with aberrant or irregular mitochondria or an equivalent

Line 142: delete “in”

Line 329: define BPA

Line: 392: see above Line 69

Reply:

All the mistakes have been corrected in the revised manuscript.

Reference

Barker N. et al. Crypt stem cells as the cells-of-origin of intestinal cancer. *Nature*. **457**, 608-611 (2009).

Schuijers J. et al. Robust cre-mediated recombination in small intestinal stem cells utilizing the *olm4* locus. *Stem Cell Reports*. **3**, 234-241 (2014).

Wang B. et al. Self-renewing diploid Axin2(+) cells fuel homeostatic renewal of the liver. *Nature*. **524**, 180-185 (2015).

Langowski JL. et al. IL-23 promotes tumour incidence and growth. *Nature*. **442**, 461-465 (2006).

Sagiv-Barfi I. et al. Eradication of spontaneous malignancy by local immunotherapy. *Sci Transl Med*. **10**, pii: eaan4488 (2018).

Klonisch T. et al. Cancer stem cell markers in common cancers - therapeutic implications. *Trends Mol Med*. **14**, 450-460 (2008).

Gordillo M. et al. Orchestrating liver development. *Development*. **142**, 2094–2108 (2015).

Wang Y. et al. Regulation of PD-L1: Emerging Routes for Targeting Tumor Immune Evasion. *Front Pharmacol*. **9**, 536 (2018).

Zerdes I. et al. Genetic, transcriptional and post-translational regulation of the programmed death protein ligand 1 in cancer: biology and clinical correlations. *Oncogene*. **37**, 4639–4661 (2018).

REVIEWERS' COMMENTS:

Reviewer #1 (Remarks to the Author):

I consider that the authors have adequately revised and extended the manuscript, the results are very interesting and will definitely benefit the further research in the field. The current manuscript is worth to be published in Nature Communication.

Reviewer #2 (Remarks to the Author):

The comments have been well addressed. It is surprising to see the in-vitro cultured cell lines contains high percentage of cancer stem cells. It would be interested to see more discussion in the rebuttal, not necessary in the manuscript though.

Reviewer #3 (Remarks to the Author):

The revised manuscript has been improved with the authors additional experiments and I can now support its publication in Nature Communications. I commend the authors on their body of work.

REVIEWERS' COMMENTS:

Reviewer #1 (Remarks to the Author):

I consider that the authors have adequately revised and extended the manuscript, the results are very interesting and will definitely benefit the further research in the field. The current manuscript is worth to be published in Nature Communication.

Reply: We thank the reviewer for positive comments on our work.

Reviewer #2 (Remarks to the Author):

The comments have been well addressed. It is surprising to see the in-vitro cultured cell lines contains high percentage of cancer stem cells. It would be interested to see more discussion in the rebuttal, not necessary in the manuscript though.

Reply: We thank the reviewer for positive comments on our work. And we also surprising that there are a lot of cancer stem cells in the in vitro cell line, such as CT26, 4T1 and B16-F10. All these three cell lines can grow transplant tumor in the wild type mice. Maybe the high cancer stem cell levels that can make them escape the immune system in the body, and subcutaneously grow transplant tumor in the balb/c or C57BL6 mice.

Reviewer #3 (Remarks to the Author):

The revised manuscript has been improved with the authors additional experiments and I can now support its publication in Nature Communications. I commend the authors on their body of work.

Reply: We thank the reviewer for positive comments on our work.